# UIS-Digger: Towards Comprehensive Research Agent Systems for Real-world Unindexed Information Seeking

**Chang Liu**[1]*, **Chuqiao Kuang**[1]*, **Tianyi Zhuang**[1]*,
**Yuxin Cheng**[2], **Huichi Zhou**[3], **Xiaoguang Li**[1], **Lifeng Shang**[1]

[1]Huawei Technologies Ltd.   [2]The University of Hong Kong   [3]University College London
{LiuChang730, kuangchuqiao, zhuangtianyi}@huawei.com
yxcheng@connect.hku.hk  huichi.Zhou.25@ucl.ac.uk
{lixiaoguang11, Shang.Lifeng}@huawei.com

## Abstract

Recent advancements in LLM-based information-seeking agents have achieved record-breaking performance on established benchmarks. However, these agents remain heavily reliant on search-engine-indexed knowledge, leaving a critical blind spot: Unindexed Information Seeking (UIS). This paper identifies and explores the UIS problem, where vital information is not captured by search engine crawlers, such as overlooked content, dynamic webpages, and embedded files. Despite its significance, UIS remains an underexplored challenge. To address this gap, we introduce UIS-QA, the first dedicated UIS benchmark, comprising 110 expert-annotated QA pairs. Notably, even state-of-the-art agents experience a drastic performance drop on UIS-QA (e.g., from 70.90 on GAIA and 46.70 on BrowseComp-zh to 24.55 on UIS-QA), underscoring the severity of the problem. To mitigate this, we propose UIS-Digger, a novel multi-agent framework that incorporates dual-mode browsing and enables simultaneous webpage searching and file parsing. With a relatively small ∼30B-parameter backbone LLM optimized using SFT and RFT training strategies, UIS-Digger sets a strong baseline at 27.27%, outperforming systems integrating sophisticated LLMs such as O3 and GPT-4.1. This demonstrates the importance of proactive interaction with unindexed sources for effective and comprehensive information-seeking. Our work not only uncovers a fundamental limitation in current agent evaluation paradigms but also provides the first toolkit for advancing UIS research, defining a new and promising direction for robust information-seeking systems.

## 1 Introduction

With the emergence of Large Language Models (LLMs) augmented by tool calls and agent-based workflow designs, modern AI systems have demonstrated impressive capabilities in performing complex real-world information seeking tasks (OpenAI, 2025). These methods usually leverage powerful tools such as search engines and crawlers for retrieving external knowledge (Team, 2025b;a; Li et al., 2025a), which we term as *Indexed Information Seeking (IIS)*. While existing benchmarks such as GAIA (Mialon et al., 2023a) and BrowseComp (OpenAI Team, 2025) suggest current agent system's advancement in information seeking, these benchmarks do not explicitly measure the extend of agent's reliance on search engines or the ability in discovering information scattered across unindexed pages.

In real-world scenarios, however, many tasks involve *unindexed information seeking (UIS)*, where necessary information is hidden in obscure corners of the Internet, embedded in files, or excluded from search engine indices due to crawling and ranking limitations. As shown in Fig. 1, for UIS questions, search engines may return related pages but fail to provide the direct content needed and interactions such as date selection and visual graph reading. Recognizing that existing benchmarks

---

*Equal contribution.

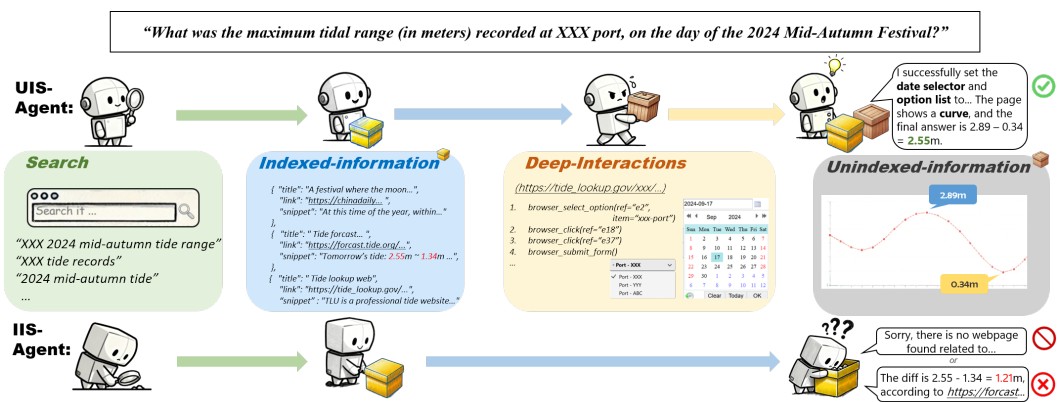

Figure 1: UIS problem. Previous information-seeking agents (bottom) focus primarily on indexed information and thus often fail to gather the evidence needed to answer complex queries, either rejecting to answer or generate hallucinations. In contrast, UIS agents (top) are equipped with additional tools and fine-tuned to excavate unindexed information, thus capable of interacting with websites deeply and solve UIS tasks reliably.

overlook the intrinsic distinction between IIS and UIS, which leads to insufficient adaptation and evaluation on agent's UIS capability, we introduce **UIS-QA**, the first benchmark explicitly designed for UIS capability evaluation. It consists of 110 carefully annotated and cross-validated test samples, ensuring correctness, objectivity, and temporal stability. The tasks in UIS-QA cover a wide range of action spaces including search, crawl page, download files and webpage interaction (e.g., select option), requiring agents to skillfully interact with webpages and excavate unindexed information.

Our experiment results reveal that even the top information-seeking agents struggle in UIS-QA, where the strongest baseline yielding only 25.45% accuracy, much lower than their GAIA and BrowseComp-zh (Zhou et al., 2025b) performance (over 70% and 45%, respectively). This findings highlight the significance of an UIS benchmark towards comprehensive evaluation of information seeking agents. With comprehensive analysis on the failure modes, we also identify two major causes of methods that perform poorly on UIS tasks, namely, the insufficient action space and limited foundation models.

As mentioned above, solving UIS tasks usually involved a wide range of actions, which can be out of the action space for search-engine-based agents (Li et al., 2025a; Team, 2025b; openPanGu Team, 2025; Shi et al., 2025), making UIS problems theoretically unsolvable. While the action space affecting the upper bound, the capability of foundation models sets the lower bound, determining whether the agent can take correct choices and forms a rational strategy within the large action space.

To this end, we also propose **UIS-Digger**, a multi-agent system for deep research tasks, with a versatile framework and a tuned inner LLM, supporting affluent actions of searching and deep browsing. The dual-mode browser within UIS-Digger allows the agent to dynamically switch between visual (screenshot) and textual modes, providing richer and more efficient webpage understanding. Furthermore, UIS-Digger also incorporates file readers and parallel tool execution, significantly strengthening its UIS-solving ability. We also tuned the underlying LLM using synthesized QA pairs through two stages: an initial supervised fine-tuning (SFT) round for cold start, followed by rejection sampling fine-tuning (RFT) for bootstrapping UIS capability. The final system achieves 27.27% accuracy on UIS-QA using ∼30B backbone LLMs, surpassing all existing baselines, including those integrating sophisticated LLMs such as O3 and GPT-4.1.

We summarize the principal contributions of this work as follows:

- We identify and formalize the overlooked problem of **Unindexed Information Seeking (UIS)**, highlighting its intrinsic distinction from IIS and demonstrating that even state-of-the-art information-seeking agents remain limited in UIS scenarios.

- We introduce **UIS-QA**, the first benchmark dedicated to UIS, featuring a rigorously validated dataset for systematically evaluating agent systems. Alongside, we propose **UIS-**

**Digger**, a versatile multi-agent framework that serves as a strong baseline, achieving a best-in-class score of 27.27%.

- We conduct detailed analyses of failure cases and agent behavior evolution across training stages, offering concrete insights and resources to guide future research in advancing the UIS domain.

## 2 UIS-QA

Since there are scarce previous studies explored the UIS problem, how to evaluate an agent system's ability under UIS task setting is still a missing piece of the puzzle. Therefore, we propose a new benchmark named UIS-QA. In this section, we will elaborate the construction of UIS-QA in three parts: the problem formulation, the data collection procedure, and the UIS filtering.

### 2.1 PROBLEM DEFINITION

To begin with, the whole Internet can be understand as a structured collection of a vast number of webpages $\mathcal{P}$. As one of the most prevalent entrances of the Internet, search engines $\mathcal{E}$ normally have crawled and organized a large portion of the webpages, denoted as $\mathcal{P}^{(\mathcal{E})}$, which is also known as 'indexed pages'. The information retrieved by a search engine is thus defined as 'indexed information'. We formalize this concept as follows:

$$\mathcal{P}^{(\mathcal{E})} = \{p_i\} = \{(u_i, s_i) \mid u_i \in \mathbf{u}, s_i \in \mathbf{s}\} \tag{1}$$

$$\mathcal{II} = \{x \mid x \in \mathbf{s} \cup crawl(\mathbf{u})\}, \tag{2}$$

where each webpage retrieved by the search engine is represented as a tuple of a URL $u_i$ and a snippet $s_i$. The collections of all the URLs and snippets from $\mathcal{P}^{(\mathcal{E})}$ are represented as $\mathbf{s}$ and $\mathbf{u}$, respectively. In other words, all the information present in the page snippets or in the results of one-step crawling from indexed pages can be considered as indexed information. Unless specified otherwise, in the following sections we use Google Serper[1] as the default search engine for $\mathcal{E}$.

Conversely, 'unindexed information' refers to all other information on the Internet excluded $\mathcal{II}$:

$$\mathcal{UI} = \{x \mid x \in (\mathcal{P} \setminus \mathcal{II})\} \tag{3}$$

In practical terms, it is infeasible to examine all the pages indexed by a search engine. Thus, the above definition serves as a theoretical model. In reality, due to computational constraints, only a small set of search queries can be fed into the search engine and only a few top pages returned from the searches will have chance to be visited. Hence, we introduce approximations for $\mathcal{II}$ and $\mathcal{UI}$:

$$\mathcal{A}(\mathcal{Q}) \rightsquigarrow \tilde{\mathcal{P}} = \{\mathcal{E}(q_i)\}^{i=1,2,\cdots,m} = \{(\tilde{u}_j, \tilde{s}_j)\} \tag{4}$$

$$\tilde{\mathcal{II}} = \{x \mid x \in (\tilde{\mathbf{s}} \cup crawl(\tilde{\mathbf{u}})\} \tag{5}$$

$$\tilde{\mathcal{UI}} = \{x \mid x \in \mathcal{P} \setminus \tilde{\mathcal{II}}\} \tag{6}$$

Here, $\mathcal{A}$ denotes an arbitrary information-seeking agent system that receives a task $\mathcal{Q}$ from the user and formulates $m$ search queries $q_i$ for searching via $\mathcal{E}$. $\tilde{\mathcal{II}}$ represents the practically accessible indexed information based on the search engine $\mathcal{E}$ and queries $\{q_i\}$, which is a subset of the ideal $\mathcal{II}$. Consequently, the remainder of $\mathcal{P}$ not included in $\tilde{\mathcal{II}}$ becomes unindexed information $\tilde{\mathcal{UI}}$.

Compared to the ideal definition, in practice, $\tilde{\mathcal{II}}$ is much smaller than $\mathcal{II}$, making it more likely that the target information necessary to solve the user's task is located in $\tilde{\mathcal{UI}}$. This practical limitation highlights the widespread and critical nature of the UIS problem.

Based on the definition of unindexed information, we further formalize the UIS problem as follow:

$$(\mathcal{Q}, \mathcal{C}) \Rightarrow z, \tag{7}$$

$$\mathcal{C} = \mathcal{C}^{(I)} \cup \mathcal{C}^{(U)} = \{c \mid c \in \tilde{\mathcal{II}}\} \cup \{c \mid c \in \tilde{\mathcal{UI}}\}, \tag{8}$$

$$where \ |\mathcal{C}^{(U)}| > 0, \ and \ (\mathcal{Q}, \mathcal{C}^{(I)}) \not\Rightarrow z \tag{9}$$

$$\tag{10}$$

---

[1]https://www.serper.dev

| | Task Type | Real-Wold Web Environment | Unknown Startpoint | Unindexed-Information Dependence | Final Answer-oriented Evaluation |
|---|---|---|---|---|---|
| WebArena | Computer Use | ✗ | ✗ | - | ✗ |
| Mind2Web | Computer Use | ✗ | ✗ | - | ✗ |
| Mind2Web-Live | Computer Use | ✓ | ✗ | ✓ | ✗ |
| Online-Mind2Web | Computer Use | ✓ | ✗ | ✓ | ✗ |
| Browsecomp-en/zh | Info Seeking | ✓ | ✓ | ✗ | ✓ |
| xbench-DeepSearch | Info Seeking | ✓ | ✓ | ✗ | ✓ |
| GAIA-textual-103 | Info Seeking | ✓ | ✓ | ✗ | ✓ |
| **UIS-QA** | Info Seeking | ✓ | ✓ | ✓ | ✓ |

Table 1: Comparison of our UIS-QA and existing benchmarks.

To solve a user's question $\mathcal{Q}$, a context $\mathcal{C}$ consisting of both indexed and unindexed information is required, denoted as $\mathcal{C}^{(I)}$ and $\mathcal{C}^{(U)}$, respectively. If the required unindexed information is not empty and the correct answer $z$ cannot be inferred from $\mathcal{C}^{(I)}$, then the $\mathcal{Q}$ is a UIS problem.

We compare UIS-QA with existing information-seeking and computer-use datasets along five key dimensions, as summarized in Tab. 1. **Task Type:** Information-seeking datasets (e.g., Browsecomp-en/zh (OpenAI Team, 2025; Zhou et al., 2025b), xbench-DeepSearch(Chen et al., 2025b), GAIA-textual-103(Mialon et al., 2023b)) require multi-step exploration on the open web, emphasizing search strategy and information extraction. In contrast, computer-use datasets (e.g., WebArena(Zhou et al., 2024), Mind2Web(Deng et al., 2023), Mind2Web-Live(Pan et al., 2024), Online-Mind2Web(Xue et al., 2025)) focus on performing interactive browser actions (e.g., click, type) to accomplish user goals, prioritizing tool operation proficiency. **Real-World Web Environment:** UIS-QA evaluates in the live public Internet. This exposes agents to real-world complexities such as outdated information, distracting content, complex layouts, and advertisements—challenges largely absent in controlled settings. **Unknown Startpoint:** UIS-QA provides no predefined starting point. Agents must initiate searches using general-purpose engines (e.g., Google) and navigate the entire web, without being restricted to specific sites. **Unindexed-Information Dependence:** UIS-QA uniquely requires reliance on information not directly accessible via standard search results. **Final Answer-Oriented Evaluation:** The benchmark employs deterministic short-form answers for evaluation, minimizing subjective judgment and enabling fully automatic scoring. In summary, UIS-QA holistically evaluates the integration of information-seeking and computer-use capabilities under realistic and demanding web interaction settings.

## 2.2 Data Collection

Following the definition of UIS tasks, we form an expert group to manually annotate question-answer (QA) pairs, and filter out those can be solved using only indexed-information solely. This process resulted in a test set of 110 high-quality UIS data samples. Specifically, the team is asked to navigate deeply authoritative or official websites, performing interactive actions such as multi-round clicks, option list selection, setting filters, intra-searching, and downloading files. Afterward, the annotators arrive at an information source such as a specific webpage or file. Based on the content of this page or file, the annotator then formulated a question, whose answer could be found or inferred from the available content. For each website, we restrict the annotators to compose a maximum of two QA pairs to ensure diversity. To further improve the quality of the annotation process, we emphasized the following principles:

*Objectivity:* unlike open-ended or subjective questions, our setting requires answers in the form of factual fill-in-the-blank questions. Thus, the answer $z$ to each question $\mathcal{Q}$ is expected to be objective, deterministic, and unique.

*Authoritativeness:* our golden answers are strictly derived from authoritative sources. Due to the intrinsic nature of UIS, such sources are often not searchable and demand strong world modeling ability to know which websites contain the appropriate authoritative information. This challenges the model to identify reliable sources amid abundant secondary and conflicting information.

*Static Nature:* given the dynamic nature of the internet, some content may change significantly over time (e.g., "What is today's weather?"), making it unsuitable for our benchmark. Therefore,

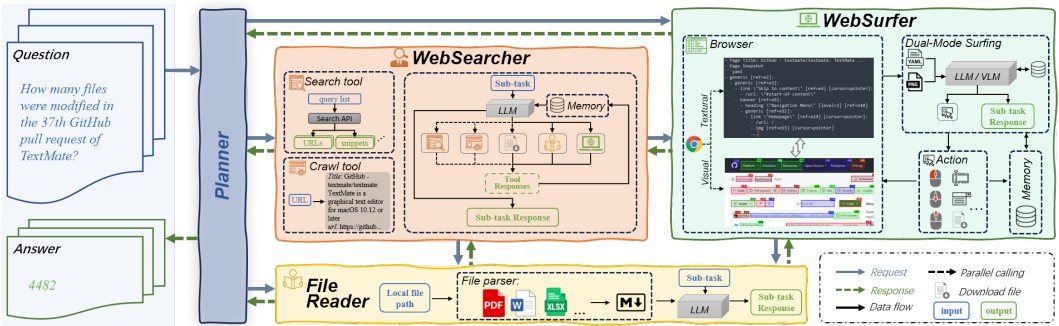

Figure 2: UIS-Digger multi-agent system. Planner, web searcher, web surfer and file reader works together to solve UIS problems. The web surfer can switch between textual- and visual-mode to observe webpages and hence make next-step decisions. Zoom-in for better view.

annotators were instructed to ensure that answers are static, so that the comparisons between agents could be fair across different testing times.

*Verifiability:* to assess the performance of agent systems on UIS-QA, we use a rule-based LLM as a verification tool. Consequently, the answers must be verifiable. Most of the "golden" answers are presented in the form of numerical values, dates, logical statements, or proper nouns. Some answers are also defined by unambiguous rules (e.g., "including either A or B can be considered correct").

*Accessibility:* annotators are asked to avoid posing questions that would trigger human verification (e.g., CAPCHAs) during the browing process. Similarly, websites with access-restricted content requiring a login are also excluded from consideration.

## 2.3 UIS FILTERING

Even under the strict collection rules described above, some questions are still inevitably solvable using only indexed information. Therefore, we design a UIS filtering pipeline to remove IIS questions. Firstly, each question is independently examined by three annotators, who use Google Search to check whether the target content can be directly retrieved from the search engine. If the search engine result page does not directly contain the target content but contains a link that redirects to the actual content page, the question is still considered UIS. In addition to manual verification, we employ z.ai[2] as an automatic verifier to filter out IIS questions. However, if a question can be answered by z.ai only after downloading a file, we classify it as UIS, since file access requires explicit browsing actions beyond indexed snippets. Next, we leverage an offline LLM (e.g., Deepseek-R1) to filter out questions answerable from LLM's inner knowledge (DeepSeek-AI, 2025). Finally, we obtain 110 high-quality samples that constitute UIS-QA.

Among the 110 samples in UIS-QA, 84 questions are written in Chinese and the remaining in English. The questions span a variety of domains, including government announcements, official product introductions, source code repositories, games, and company annual reports.

## 3 UIS-DIGGER: A MULTI-AGENT FRAMEWORK FOR UIS PROBLEMS

As mentioned above, there are few existing works that have studied the UIS problem. Therefore, we propose a new agent-system framework for UIS problem solving, named UIS-Digger, which can serve as a fundamental methodology for UIS-QA. In this section, UIS-Digger will be elaborated in three aspects of agent design, framework architecture, and training process.

### 3.1 AGENT AND ARCHITECTURE DESIGN

In Fig. 2 we introduce the overall architecture of UIS-Digger. UIS-Digger is a multi-agent system engaging four agents, planner, web searcher, web surfer and file reader. Each agent is equipped

---

[2]https://chat.z.ai/

with a set of tools and assigned to a specific category of sub tasks. For every new instruction, the agent initializes an empty memory and works in an iterative problem-solving process inspired by the ReAct paradigm (Yao et al., 2023). The agents communicate with each other and corresponding tools via a request-response message system.

*Planner:* upon receiving a new user query, the top-level planner decomposes it into a set of sub-tasks, coordinates the execution among the three subordinate agents, and delivers the final answer to the user.

*Web Searcher:* the web searcher concurrently employs search engines and crawling tools to retrieve indexed information (Eq. 5), and may further delegate sub-tasks to the web surfer and file reader to obtain unindexed information from web URLs or files.

*Web Surfer:* The web surfer starts from a URL and operates a browser to access unindexed information. Its action space covers common interactions with websites, including clicking, scrolling, typing, selecting, navigating, submitting forms, downloading files, locating elements, and taking screenshots. Unlike previous browser-integrated methods with either purely textual or visual observation about the webpage (Zheng et al., 2025; CAMEL-AI.org, 2025), we introduce a dual-model memory-shared browsing strategy, to balance both completeness in functionality and high efficiency. Crucially, unlike previous multimodal agents, our surfer maintains a shared memory and consistent browser state across textual and visual modes. This design preserves a unified working history, eliminates synchronization overhead, and encourages efficient inference by prioritizing textual mode while reserving visual inspection for essential cases.

*File Reader:* both the information seeker and web surfer can download files, which are then processed by the file reader supporting formats such as PDF, XLSX, and DOCX. When content exceeds the context window, it is incrementally read chunk by chunk, following Yu et al. (2025b).

## 3.2 AGENT TRAINING

UIS-Digger requires specialized capabilities from its inner LLMs, including task decomposition, tool usage, and integrating diverse information for UIS tasks. To this end, we construct synthesized training data and tune the inner LLMs in two stages of SFT and RFT.

### 3.2.1 TRAINING DATA CONSTRUCTION

For efficiency, we synthesize QA pairs rather than rely solely on manual annotation. We draw upon both real-world information from the internet and simulated environments, as illustrated in Fig. 3.

To construct QA pairs from real-world information sources, over one hundred base websites are collected, across domains such as public companies, product catalogs, government announcements, data dashboards, and code repositories. UIS-Digger is instructed to roam within these websites and extract five informative sections about a chosen entity, forming a context as defined in Eq. 8. It is designed to gather information from deeper webpages by performing various browsing actions. Then we deploy another LLM to compose a question and label the corresponding answer based on this context, followed by an LLM judge filtering out ambiguous or subjective questions. The prompts used for information collection and query generation are provided in Appendix B.

To address early weaknesses in handling interactive web elements, such as selecting a date in a datetime picker, we further developed three types of virtual websites that simulate flight booking and statistical data lookup scenarios. These websites incorporate specific interactive elements that posed challenges to the earlier version of UIS-Digger. Each virtual site is provided a fictitious JSON database (e.g., synthetic shopping records). QA pairs can be directly derived from the database, while UIS-Digger must solve them by interacting with the simulated website. This simulation strategy significantly enhances the agent's ability to manipulate widgets such as radio buttons, date selectors, filters, and graphs.

Based on the constructed QA pairs from real and virtual websites, we employ UIS-Digger to solve these questions and collect the trajectories, which are then filtered with reject-sampling method and used for tuning the inner LLM of UIS-Digger. The final result trajectories are used in two stages of SFT and RFT, with disjoint question sets allocated to each.

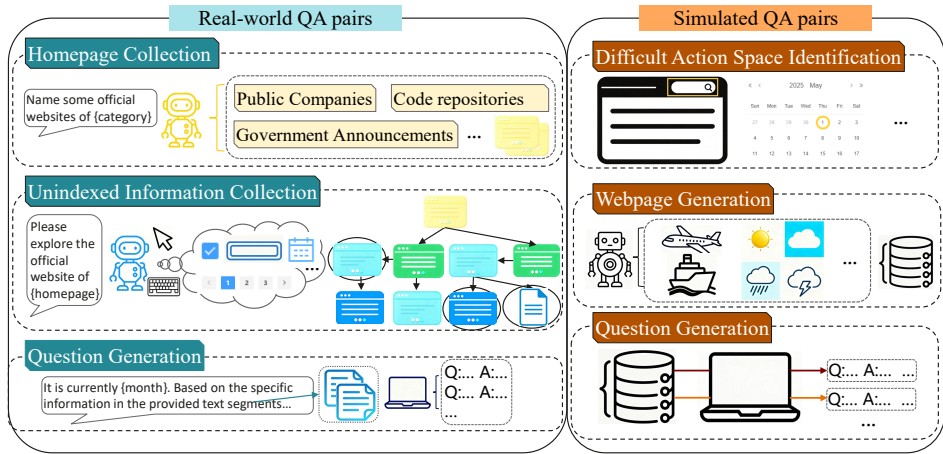

Figure 3: QA Pairs Construction Pipeline. (**Left**): Procedure for constructing QA pairs using real-world information. First, homepages potentially containing deep navigation structures and informative content are collected. UIS-Digger then explores these homepages to extract information from pages requiring multiple navigation steps. The collected information subsequently serves as context for query generation. (**Right**): Procedure for constructing QA pairs based on simulated webpages. We identify browsing actions that UIS-Digger struggles to perform, and generate webpages (along with a JSON database containing relevant statistics) that incorporate these actions. QA pairs are then generated using the information from the JSON database of these simulated webpages.

### 3.2.2 Two-stage Training

In the SFT stage, we integrate a powerful teacher model $\mathcal{X}^*$ to solve sampled questions with temperature 0, producing one trajectory per question. A separate LLM judge verifies (1) the correctness of the final answer and (2) whether the question is trivial, i.e., if first-round reply already contains the golden answer $z$. We adopt reject sampling, retaining only those correct and non-trivial trajectories. The resulting SFT-tuned model is denoted as $\mathcal{X}^s$, which is then used for RFT trajectory generation.

In the RFT stage, $\mathcal{X}^s$ is deployed to solve the remaining training questions, with temperature 0.4 and a sampling group size of four, encouraging exploration. The same reject-sampling strategy is applied. To emphasize challenging tasks, we reweight samples by difficulty, measured by the number of correct attempts. Specifically, trajectories from challenging questions are more likely to be retained than those from easier ones. Bootstrapping with these RFT trajectories yields the final model $\mathcal{X}^r$, which is integrated as the default LLM in UIS-Digger. Unless otherwise specified, all subsequent experimental results are reported with $\mathcal{X}^r$.

## 4 Experiments

In this section, we present results and analyses on the proposed benchmark UIS-QA and the UIS-Digger system. Across nearly all baseline methods, we observe substantial performance gaps between UIS-QA and prior non-UIS tasks, underscoring the significance of UIS-QA as a novel benchmark. Furthermore, through error case analysis across different models, we identify several key factors that determine an agent system's success in UIS.

### 4.1 Experimental Settings

To distinguish the different nature of the proposed UIS-QA benchmark from exisiting ones, we conduct affluent evaluations on existing advanced information seeking agents:

- **Direct API Inference** These methods directly query a base LLM through provider's APIs, with action space (e.g., whether can use search tools) not fully disclosed. We evaluate models such as DeepSeek-V3.1 (DeepSeek-AI, 2024), Claude-sonnet-4 (Anthropic, 2025) and GPT-5 (OpenAI, 2025).

| Names | Action Space | | | | Backbone | UIS-QA | GAIA | BC-zh |
|---|---|---|---|---|---|---|---|---|
| | crawl(s) | visual | file | browser | | | | |
| *Direct Inference* | | | | | | | | |
| DeepSeek-V3.1 | - | - | - | - | DeepSeek-V3.1 | 1.8 | - | - |
| Claude-sonnet-4 | - | - | - | - | Claude-S4 | 2.7 | - | - |
| GPT-5 | - | - | - | - | GPT-5 | 0.9 | - | - |
| *Commercial System* | | | | | | | | |
| GLM-4.5 auto-thinking, web search | ✓ | ✗ | - | ✓ | GLM4.5† | 11.8 | - | - |
| Doubao DeepThink | - | - | - | - | Doubao | 11.8 | - | - |
| Gemini-2.5-pro google_search | - | - | - | - | Gemini-2.5-pro | 4.5 | - | - |
| *ReAct Agentic Framework* | | | | | | | | |
| WebSailor | ✓ | ✗ | ✗ | ✗ | WebSailor-32B + Qwen3-72B | 7.3 | 53.2‡ | 25.5 |
| Tongyi-DR | ✓ | ✗ | ✗ | ✗ | TongyiDR-30B-A3B†+GPT-4o | 23.6 | 70.9‡ | **46.7** |
| *Multi-agent Framework* | | | | | | | | |
| DDv2 | ✓ | ✗ | ✗ | ✗ | Pangu-38B | 8.2 | - | *34.6* |
| OWL | ✓ | ✓ | ✓ | ✓ | O3-mini + 4o + Claude-S3.7 | 4.6 | 69.7 | - |
| MiroThinker v0.1 | ✓ | ✓ | ✓ | ✓ | MiroThinker-32B-DPO + GPT-4.1 +Claude-S3.7 | 7.3 | 57.9‡ | |
| Memento | ✓ | ✓ | ✓ | ✗ | O3 + GPT-4.1 | 25.5§ | **79.4** | - |
| AWorld | ✓ | ✓ | ✓ | ✓ | Gemini-2.5-pro + GPT-4o | 5.5 | 32.2 | - |
| UIS-Digger (Pangu) | ✓ | ✓ | ✓ | ✓ | PanGu-38B | **27.3** | 50.5 | 32.5 |
| UIS-Digger (Qwen) | ✓ | ✓ | ✓ | ✓ | Qwen3-32B | **27.3** | 47.6 | 32.5 |

Table 2: Evaluation results on UIS-QA, GAIA, and BrowseComp-zh (BC-zh). † indicates reasoning-oriented LLMs. ‡ denotes results measured on GAIA-text-103 rather than the full GAIA benchmark. § indicates that the UIS-QA score for Memento (Zhou et al., 2025a) is reported without using its case bank, since UIS is a new task type and only limited cases have been previously allocated. Action spaces including crawl (read webpage content), visual (read images), download file and opearte browser are included.

- **Commercial Systems.** Beyond a single LLM, these systems adopt more sophisticated architectures that theoretically enable a broader action space such as searching. GLM-4.5 (Team et al., 2025), Doubao (Seed team, 2025), Gemini-2.5-pro (DeepMind, 2025) belongs to this category.

- **ReAct-based Frameworks.** A straightforward agent design that couples reasoning and action, represented by WebSailor (Li et al., 2025a) and Tongyi DeepResearch (Team, 2025b).

- **Multi-agent Frameworks.** These methods implement multi-agent architectures where specialized agents handle different tasks such as webpage crawling, visual signal interpretation, file reading, and browser operation. Many systems in this group achieve strong results on traditional benchmarks like GAIA and BrowseComp. Examples include DDv2 (open-PanGu Team, 2025), OWL (CAMEL-AI.org, 2025), MiroThinker (Team, 2025a), Memento (Zhou et al., 2025a), and AWorld (Yu et al., 2025a).

The proposed UIS-Digger with backbone $\mathcal{X}^r$ is also evaluated in this part. We trained two versions of $\mathcal{X}^r$, a 38B-Pangu model (Chen et al., 2025a) and a Qwen3-32B model (Yang et al., 2025). During training, only LLM-generated tokens are updated with gradient backpropagation, while tool responses are excluded. Implementation details of the two stages are provided in Appendix C.

## 4.2 MAIN RESULTS ON UIS-QA

In Tab. 2, we present the evaluation results of baseline methods and UIS-Digger. UIS-Digger achieves the highest score of 27.27% on the UIS-QAbenchmark, outperforming even sophisticated systems powered by O3. In addition, it delivers competitive results on conventional information-seeking benchmarks such as GAIA and BC-zh, demonstrating strong generality. These findings suggest that UIS-Digger establishes a solid baseline for advancing research on the UIS problem.

By contrast, all baseline methods suffer substantial accuracy drops under the UIS setting. Even strong systems such as Tongyi-DR and Memento, which exceed 70% accuracy on GAIA, drop to only 23.6% and 25.5% on UIS-QA—corresponding to declines of 47.3% and 53.9%, respectively. This sharp degradation reinforces our central motivation: UIS remains an underexplored and insufficiently addressed capability in current agent systems.

Beyond the ranking of baseline methods, it is also worthy to note that methods that achieve higher scores on general information-seeking tasks such as GAIA also tend to perform relatively better on UIS-QA. This correlation suggests that a strong foundation model (e.g., O3 in Memento) is still essential for UIS tasks.

Nevertheless, When comparing ReAct-style methods with more complex agent frameworks, we observe that the relative distribution of UIS and IIS scores is not fundamentally different. Even within the same framework type and similar action spaces, these methods exhibit large performance disparities, with gaps of up to 17.3% and 20.9%, respectively. We hypothesize that while a larger action space theoretically enables more diverse strategies, it also expands the search space and introduces new challenges. The main bottleneck, therefore, shifts to the underlying LLM's fundamental ability.

## 4.3 ANALYSIS

To systematically analyze the challenges faced by agent systems in solving UIS tasks, we conduct a detailed examination of their searching and browsing behaviors. Fig. 4 illustrates two key aspects: the proportion of trials successfully grounded to the golden information source (left), and the action frequency distributions across correct and incorrect samples after different training stages (right).

**Gains from SFT and RFT Training**  Both SFT and RFT training stages lead to substantial accuracy improvements on UIS-QA, demonstrating the effectiveness of the two-stage tuning strategy. For instance, UIS-Digger with a PanGu backbone achieves gains of 13.6% from SFT and an additional 4.6% from RFT. Further details and extended results are provided in Appendix D.1.

**Error Analysis**  On the left side of Fig. 4, we analyze the searching behaviors of four representative methods—Memento, Tongyi-DR, WebSailor, and UIS-Digger—on UIS-QA. We evaluate whether an agent successfully retrieves and accesses the root website of the annotated golden information source. The root website is defined as the domain name of the ground-truth webpage, and actions such as crawling, surfing, or downloading the golden webpage URL are counted as visits. The three concentric rings in each pie chart, from the innermost to the outermost, denote: (1) final answer correctness, (2) whether the golden root website is retrieved during search, and (3) whether the golden root website is subsequently accessed.

Observed from different parts of the pie charts, we identify several key patterns. For clarity, sections are denoted by their colors from the inner to outer rings (e.g., BBR stands for Blue–Blue–Red). More illustrative examples are provided in the Appendix E.

*Missing retrieval (RRR) and knowledge sourcing (RBR) are two dominant failure modes.* Without retrieving the root page, solving a UIS problem becomes theoretically impossible, underscoring the need for robust search capabilities. Even when homepages are retrieved, agents often fail to select the correct knowledge source among the results, highlighting the importance of precise source identification. These patterns emphasize the value of UIS-QAin exposing UIS-specific weaknesses in agent behaviors.

*UIS remains difficult even when the source page is reached (RBB).* Another substantial fraction of cases involve correctly retrieving and visiting the root website but still producing incorrect final answers. Such failures stem from the inherent complexity of UIS action spaces: even when starting from the correct source, agents must execute intricate operation sequences—such as multi-step navigation, filter adjustments, or repeated back-and-forth exploration. This calls for stronger continuous reasoning and long-horizon planning capabilities in future agent systems.

*Intrinsic knowledge and alternative sources offer only limited shortcuts.* We also observe a small number of correct cases where the golden root website is neither retrieved nor visited. Our manual inspection suggests two explanations: (1) agents occasionally leverage intrinsic knowledge of URLs to directly access relevant pages, and (2) third-party websites sometimes redundantly host the required information. While such cases reveal that prior knowledge or external redundancy can occasionally "hack" UIS tasks, their rarity indicates they do not fundamentally mitigate the UIS challenge.

**Tool Usage Across Training Stages**  We observe clear shifts in tool-utilization patterns as the agent advances through training. As shown in Fig. 4 (right), the frequency of search tool calls in-

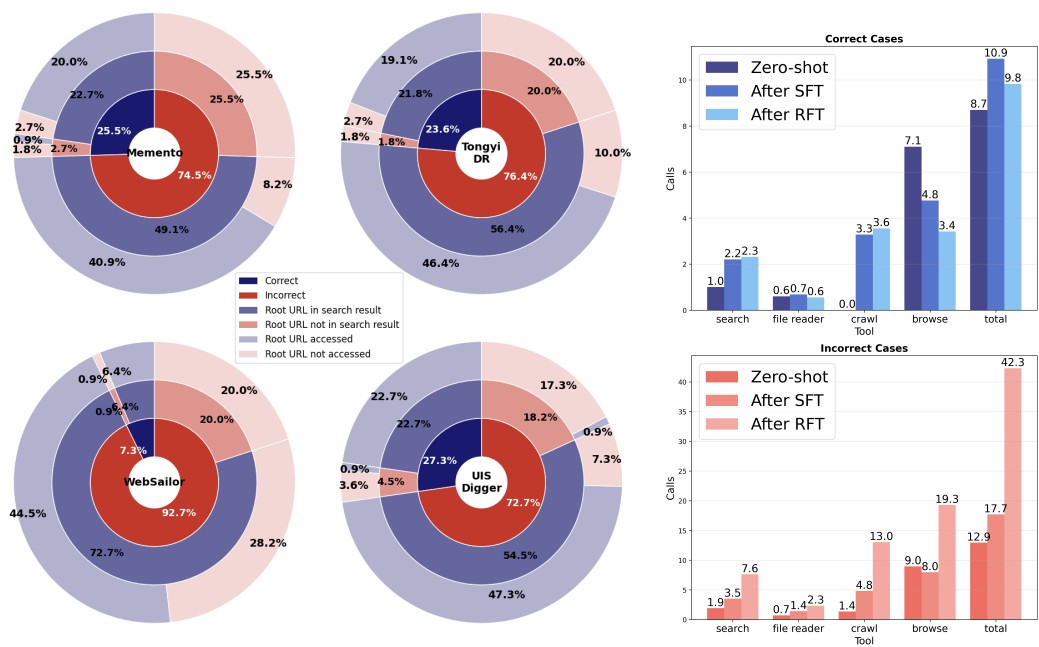

Figure 4: Action analysis. (**left**): Search behaviors of UIS-Digger and three baseline methods. The pie charts show the proportions of cases where the agent successfully retrieves the root URL via search, and whether the root URL is subsequently accessed through crawling or browsing. (**Right**): Action frequency distributions of correct and incorrect cases for Pangu-38B UIS-Digger at different training stages. Zoom-in for best view.

creases across both correct and incorrect trajectories, reflecting the growing reliance on external retrieval, which is believed to potentially reduce hallucination. In contrast, file-parsing actions remain largely unchanged, consistent with their role as a follow-up step once relevant files are downloaded.

A critical difference emerges in the use of the *crawl* tool. The untrained model fails to invoke it altogether, whereas this capability appears after SFT and further improves with RFT, underscoring the importance of staged training for acquiring essential behaviors. Browsing actions reveal another important trend: in successful trajectories, browsing attempts sharply decrease over training, indicating more targeted and efficient navigation. Conversely, unsuccessful trajectories show an increasing number of attempts, suggesting heavy unsuccessful exploration.

Overall, correct trajectories follow a trajectory of "learn then streamline": tool usage rises after SFT as the agent learns to solve more complex tasks with longer tool-use sequences, then declines as navigation efficiency improves with RFT. Incorrect trajectories, however, show a monotonic increase in tool calls, reflecting prolonged retries that fail to converge to a correct solution.

## 5 CONCLUSION

In this paper, we identify the overlooked problem of Unindexed Information Seeking (UIS), where indispensable information resides beyond the reach of search engines. To systematically evaluate this UIS capability, we introduce the UIS-QA benchmark, which provides a dedicated test set for assessing agent systems on UIS tasks. Although existing agents achieve strong performance on conventional information-seeking benchmarks, their ability to solve UIS problems remains limited. Consequently, we propose UIS-Digger, an agent system with enhanced web-interactive tools and trained through sequential SFT and RFT stages. Our results demonstrate that with an appropriate action space and tailored training strategy, UIS ability can be effectively bootstrapped, enabling UIS-Digger to achieve state-of-the-art performance on UIS-QA. Nevertheless, despite these improvements, the absolute accuracy of UIS-Digger at 27.27% remains far from satisfactory, underscoring the difficulty of UIS. We hope that UIS-QA will encourage further research in this direction and inspire the development of more practical and generalizable deep research agents.

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

## A    RELATED WORK

**Information Seeking Benchmarks**    Early benchmarks primarily focused on multi-hop question answering. For instance, HotpotQA (Yang et al., 2018) was designed to evaluate multi-hop retrieval and question answering, while SimpleQA (Wei et al., 2024) focused on short-form factual queries. Musique (Trivedi et al., 2022) further tested multi-hop reasoning over single-hop evidence.

More recent benchmarks demand deeper and more persistent search behaviors. Benchmarks such as BrowseComp-en/zh (OpenAI Team, 2025; Zhou et al., 2025b) and Blur (CH-Wang et al., 2025) incorporate deliberate information obfuscation, requiring agents to persistently navigate the web to locate hard-to-find information. Similarly, xbench-DeepSearch (Chen et al., 2025b) measures reasoning, tool usage, and memory in extended interaction chains. A common strategy among these benchmarks is to increase task difficulty by lengthening the reasoning chain, thereby necessitating multi-step browsing and tool invocation. However, they do not explicitly evaluate an agent's ability to retrieve *unindexed* information that is not easily accessible via search engines. As a result, systems excelling at conventional search may perform well on these benchmarks but fail dramatically on our proposed UIS-QA.

Other works approach complex information-seeking from different angles. WideSearch (Wong et al., 2025) evaluates broad-scale information retrieval across multiple domains and time spans, often drawing from official websites. HLE (Phan et al., 2025) focuses on challenging academic reasoning, while GAIA (Mialon et al., 2023a) emphasizes long-horizon tool use. More recently, benchmarks like WideSearch (Wong et al., 2025), FinSearchComp (Hu et al., 2025), and DeepResearch Bench (Du et al., 2025) tackle domain-specific information needs and, in doing so, occasionally involve unindexed sources through historical financial reports or specialized official data. Nevertheless, such exposure remains incidental. In contrast, our work systematically isolates **U**nindexed **I**nformation **S**eeking (UIS) as a core capability dimension, offering a principled evaluation framework.

**Information Seeking Agents**    Recent years have witnessed significant progress in the development of information-seeking agents. Technology companies have released deep research products, such as OpenAI Deep Research (OpenAI, 2025), Google Gemini Deep Research (Google, 2025), Kimi-Researcher (AI, 2025), and Grok-3 Deep Research (xAI, 2025). In parallel, the research community has explored multi-agent architectures for complex task orchestration. For example, OWL (CAMEL-AI.org, 2025) proposes a hierarchical framework of planning and specialized execution, while AWorld (Yu et al., 2025a) offers an open-source platform for large-scale agent-environment interaction.

Several studies focus on enhancing reasoning and exploration capabilities during web search. Web-Thinker (Li et al., 2025b) integrates reasoning processes with web exploration. Search-R1 (Jin et al., 2025) employs reinforcement learning to enable LLMs to autonomously generate search queries during multi-step reasoning. To address training data scarcity, methods such as SimpleDeepSearcher (Sun et al., 2025b) synthesize data by simulating realistic user interactions in live search environments, and ZeroSearch (Sun et al., 2025a) uses LLMs to simulate a search engine during training. WebDancer (Wu et al., 2025) creates challenging training tasks that demand deeper multi-hop reasoning. Furthermore, DeepDiver V2 (openPanGu Team, 2025) trains a multi-agent system on both closed-ended problems requiring extensive information gathering and verification, and open-ended tasks aimed at producing comprehensive long-form content.

To explore the unindexed information seeking capabilities of agents, we propose an agent architecture that supports flexible interaction between a planner and specialized subagents capable of directly manipulating web elements. Additionally, we enhance the backbone model through carefully curated synthetic data using both supervised fine-tuning (SFT) and rejection sampling fine-tuning (RFT).

## B  PROMPTS USED FOR QA PAIRS GENERATION

The prompts utilized for collecting information from homepage browsing and for generating the final queries are presented as follows.

---

**Prompt for Information Collection**

Please explore the official website of {homepage}. You are encouraged to conduct searches and to select in-depth pages rich in substantive content for browsing. Finally, paraphrase the content of at least **five specific articles on different topics that contain a wealth of detailed entity information**.
For example:
- Visit the Investor Relations section of a corporate website, locate the Q3 2024 report, download it, and record the shareholding percentage of the largest individual shareholder.
- Visit a museum's official website, find the "Treasures of the Museum" tab, click on it, and record all the listed treasures.
Note:
1. Paraphrase the specific content; do not use statements such as "for specific details, please refer to X document."
  1.1 You are encouraged to paraphrase information containing numerical values and names.
2. The collected content should ideally not be directly searchable via search engines.
  2.1 You are encouraged to visit related detailed pages. For example: Access "X Company's accounts payable for Q2 2025 is..." and collect the specific content, then access "X Company's accounts payable for Q1 2025 is..." and paraphrase both pieces of information.
  2.2 If documents are available on the website, you are encouraged to paraphrase the specific content within those documents.
3. The source of the specific content must be the original text you actually saw; DO NOT fabricate anything!!! Paraphrase these contents verbatim directly.
4. Select objective and specific content.
  4.1 The information provided by the content must be objective and definitive. For example: "In 2025, X's revenue rate was...", "X's standard numbers include...", "X was included in the National Patent Industrialization Demonstration Enterprise Cultivation Pool in month z of year y."
  4.2 The information provided by the content cannot be vague or allow for other possibilities. For example: "X's advantages include...", "X's main goals are...", "X focuses on aspects y and z.", "The reasons X does Y are...".
  4.3 The information provided by the content cannot be overview/summary in nature. For example: "X's key measures include...", "The difficulties in X's research include...", "X's prospects for the future include...".
  4.4 Do not select speech-type, manifesto-type, or address-type webpages.
5. Maintain rigor.
  5.1 For all content, considering the current date, version, etc., the collected content must include specific qualifying statements. Do not say "Sales of X's flagship model were y yuan"; add conditions and change it to, for example, "Sales of X's 2024 flagship model in Mainland China were y yuan". Do not say "X has a total of 41 characters"; add conditions and change it to, for example, "Version 5.2.3 of X has a total of 41 heroes".
  5.2 For content specific to a particular institution or enterprise, include the institution or enterprise as a condition. Do not say "Investment meetings are held on the last day of each quarter"; add the enterprise condition and change it to, for example, "Enterprise X's investment meetings are held on the last day of each quarter".

---

> **Prompt for Query Generation**
>
> It is currently {month}. Based on the specific information in the provided text segments, please create 5 objective questions from different perspectives that have definitive answers. Attach the answers and the rationale for each question, separating multiple rationales with semicolons. Use the format: 1. Question Design: XXX Question: XXX Answer: XXX Rationale: XXX
> Text Segment:
> {context}
> Note:
> 1. Ask questions targeting specific information; do not focus on "task" descriptions.
> 2. The questions should ideally not be answerable by directly searching a search engine.
>  2.1 You are encouraged to design multi-hop questions based on the text segment. For example: combine "What were X Company's accounts payable for Q2 2025?" and "What were X Company's accounts payable for Q1 2025?" into "By how much did X Company's accounts payable increase in Q2 2025 compared to Q1 2025?"
> 3. The source for the questions must be the original text you actually saw; DO NOT fabricate anything!!!
> 4. Ensure the objectivity and specificity of the questions.
>  4.1 A question is objective and specific if it has an objective, definitive answer. For example: "What was X's revenue rate in 2025?", "What are the standard numbers for X?", "When was X included in the National Patent Industrialization Demonstration Enterprise Cultivation Pool?"
>  4.2 A question is *not* objective and specific if the answer is open to reasonable interpretation. Avoid questions like: "What are the advantages of X?", "What are the main goals of X?", "What aspects does X focus on?", "Why does X do Y?"
>  4.3 A question is *not* objective and specific if multiple non-equivalent answers could be considered accurate. Avoid questions like: "List the key measures of X."
>  4.4 A question is *not* objective and specific if it is overview/summary in nature. Avoid questions like: "What was reported in X?", "What are the research difficulties in X?"
> 5. Maintain rigor and ensure the uniqueness of the answer.
>  5.1 For all content, considering the current date, version, etc., include specific qualifying statements. Do not ask "What were the sales of X's flagship model?"; add conditions and ask, for example, "What were the sales of X's 2024 flagship model in Mainland China?". Do not ask "How many characters does X have?"; add conditions and ask, for example, "How many heroes are in version 5.2.3 of X?"
>  5.2 For content specific to a particular institution or enterprise, include the institution or enterprise as a condition. Do not ask "On which day are investment meetings held each quarter?"; add the enterprise condition and ask, for example, "On which day of the quarter does Enterprise X hold its investment meetings?". Do not ask "on the official website"; specify which official website.
> 6. Do not include specific webpage titles or file names in the questions. The answers must not contain phrases like "for specific details, please refer to X link".

## C  IMPLEMENTATION DETAILS

**SFT Training.** For supervised fine-tuning (SFT), we use a learning rate of $3 \times 10^{-6}$ with a batch size of 32. Each training instance is packed to a sequence length of 128k tokens. We train the model for a total of 3 epochs. After filtering for correct teacher answers, we retain 1,482 training queries, corresponding to 4,501 trajectories in total. Since our framework is a multi-agent system, a single query may correspond to multiple trajectories.

**RFT Training.** For reject-sampling fine-tuning (RFT), we use the same learning rate and batch size as in SFT. After filtering for correct responses, the full RFT dataset contains 12,959 trajectories associated with 3,317 queries. After applying difficulty-weighted sampling (oversampling difficult queries and undersampling simpler ones), the final number of trajectories actually used for training is 4,467.

|            | UIS-QA | BC-zh | GAIA | FinSearchComp(T2/T3) |
| ---------- | ------ | ----- | ---- | -------------------- |
| Pangu-38B  | 9.1    | 12.1  | 25.2 | 48/3.4               |
| Pangu-SFT  | 22.7   | 30.8  | 42.7 | 69.0/5.7             |
| Pangu-RFT  | 27.3   | 32.5  | 50.5 | 73.0/11.4            |

Table 3: Performance of each training stage across different benchmarks.

## D ABLATION STUDY

In this section, we analyze the contributions of UIS-Digger's modules and technical choices. Overall, the results confirm both the robustness and the effectiveness of our framework in tackling UIS problems.

### D.1 PERFORMANCE GAINS FROM SFT AND RFT TRAINING

Beyond UIS-Digger's strong performance on UIS-QA, we conduct ablations to assess how different training stages contribute to accuracy. As shown in Fig. 5, performance consistently improves after each stage of SFT and RFT, though with diminishing returns. The most significant gain comes from the SFT stage, supporting our claim that vanilla agents lack awareness of UIS and perform poorly at the outset.

RFT further improves performance by enabling the agent to explore diverse solving strategies and reinforce successful ones. This finding is encouraging: even under the UIS setting, self-improvement through reinforcement remains effective. Nevertheless, UIS-Digger's absolute accuracy after RFT is still unsatisfactory, indicating substantial room for future works. We hypothesize two key limitations: (1) a distribution gap between synthesized QA pairs and the real test set, which weakens transfer, and (2) sparse supervision from reject sampling, where feedback is based only on final answers, potentially reinforcing low-quality trajectories. We also evaluate our method on other benchmarks to assess its generalizability. As shown in Tab. 3, consistent performance gains across various benchmarks are observed. Notably, some benchmarks exhibit even larger improvements than on UIS-QA, validating the broad effectiveness of our SFT and RFT stages.

### D.2 BACKBONE MODELS

To disentangle the impact of the backbone LLM from that of the UIS-Digger framework, we compare several models (Tab. 4). Both Pangu-38B and Qwen3-32B, when trained under UIS-Digger, achieve high score of 27.3%, demonstrating that the framework and training pipeline generalize across backbones. Similarly, Claude-sonnet-4 reaches 23.6%, showing a substantial improvement over its original performance and indicating that UIS-Digger benefits even relatively weaker backbones.

In contrast, directly deploying GPT-4o as the main LLM leads to a dramatic drop to 8.2%, while the similarly untuned O3 yield to 30.9%, which even surpass the tuned small models of Pangu and Qwen3. This finding suggests that raw foundation model capability alone is critical and compatibility with the framework can also significantly affect performance.

For the dual-mode web surfer, we also ablate the choice of VLM used to interpret visual signals. By replacing GPT-4o with QwenVL-max, UIS-Digger still achieves 25.5%, close to the original 27.3%. This demonstrates that UIS-Digger is robust to different VLM choices, with only minor performance variation.

## E CASE STUDY

This section provides detailed case analyses corresponding to the error categories discussed in Section 4.3. Each case (translated into English) illustrates a specific mode, detailing the agent's actions.

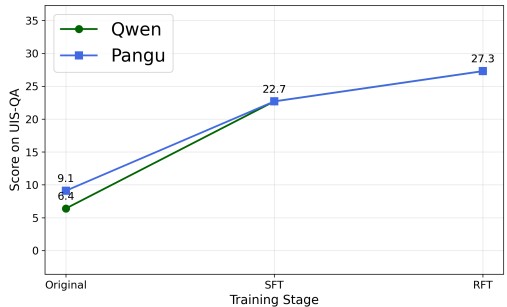

Figure 5: The UIS-QA score changing curve for UIS-Digger at different stages of training.

Table 4: Ablation results on backbone models and VLMs.

| backbone | VLM | UIS-QA |
|---|---|---|
| Pangu-38B | GPT-4o | 27.3 |
| Pangu-38B | QwenVL-max | 25.5 |
| Qwen3-32B | GPT-4o | 27.3 |
| O3-mini | GPT-4o | 7.6 |
| GPT-4o | GPT-4o | 8.2 |
| Qwen3-235b-a22b | GPT-4o | 13.9 |
| GPT-oss-120b | GPT-4o | 19.9 |
| Claude-sonnet-4 | GPT-4o | 23.6 |
| O3 | GPT-4o | 30.9 |

**Knowledge Sourcing Error**    This case demonstrates a critical failure in source selection. Despite the presence of the correct, authoritative website in the search results, the agent committed a knowledge sourcing error by opting for a third-party source. The selected source was compromised by outdated historical data that was irrelevant to the query's specific context. Consequently, the agent produced a highly confident yet incorrect answer. Blue indicates the correct source, red indicates a third-party source and orange indicates the outdated information

| | |
|---|---|
| **Question** | What committees are under the Academic Committee of the Chengdu Giant Panda Base? |
| **Gold Root Website** | www.panda.org.cn
Chengdu Research Base of Giant Panda Breeding |
| **Browsed Webpage** | clas.cas.cn
Chengdu Library and Information Center, Chinese Academy of Sciences |
| **Action** | ## Observation
Through an in-depth exploration of the website of the Chengdu Documentation and Information Center, Chinese Academy of Sciences, I have obtained detailed organizational structure information regarding the Academic Committee of the Information Service Center for the Chengdu Giant Panda Breeding Research Base. The primary sources of information include: 1. Successfully accessed the detailed page for the Academic Committee (http://clas.cas.cn/zztx2016/gltx2016/xswyh2016/) 2. Acquired the complete historical evolution and organizational structure of the Academic Committee from the 1st term (1981) to the 8th term (revised) (2019 to present)...
## Reflection & Reasoning
I have successfully completed the user's task requirements. Through systematic exploration of the website, I located an authoritative official information source and obtained complete organizational structure information for the Academic Committee of the Information Service Center for the Chengdu Giant Panda Breeding Research Base.
Key Findings:
1. The Academic Committee is an institution with a long history, established in 1981, spanning over 40 years... |

**Successful Recall of the Gold Root Website from Internal Knowledge**    This case demonstrates a scenario where the agent successfully leverages its internal knowledge of authoritative domains to navigate directly to the correct (gold root) website, bypassing the need for a potentially noisy web search.

| Question | According to the Industrial and Commercial Bank of China's RMB spot exchange rate, what was the buying price of the Euro on 2025-05-01, and what was the specific release time? |
|---|---|
| **Gold Root Website** | www.icbc.com.cn
Industrial and Commercial Bank of China |
| **LLM Input (Received from Planner)** | Please help me complete the following task. When carrying out the task, please carefully meet the requirements specified in the system prompt. Your task is: <task>Visit the official website of the Industrial and Commercial Bank of China (ICBC) and find the euro's RMB spot exchange buying rate for May 1, 2025, along with the specific release time. Please focus on checking sections such as 'Foreign Exchange Rates' or 'Financial Markets' on the ICBC website to locate the euro/RMB spot exchange buying rate data.</task> |
| **LLM Output (Tool Call)** | "task": "Visit the official website of Industrial and Commercial Bank of China (ICBC) to find the Euro to RMB spot foreign exchange buying rate for May 1, 2025, and the specific release time. Focus on checking sections like 'Foreign Exchange Rates', 'Financial Markets', or related sections on the ICBC website to locate the Euro/RMB spot forex buying price data.", "start_url": "https://www.icbc.com.cn/", "max_num_steps": 20 |

**Failure to Interact with Webpage Elements** In this case, the agent successfully navigated to the correct (gold root) website. However, it failed to complete the task due to an inability to navigate to the specific page that shows the exhibition hall map.

| Question | Which heroes in the game Honor of Kings had the highest number of appearances, ban rate, and average kills per game during the 2025 KPL Spring Season, and what were the respective figures? |
|---|---|
| **Gold Root URL** | www.chnmuseum.cn
Nation Museum of China |
| **Search Action** | "query_list": ["Number of restrooms/toilets on B1 floor of National Museum of China", "National Museum of China basement level toilets", "National Museum of China floor plan restrooms"] |
| **Search Response** | ... "title": "Partial Toilet Renovation and Upgrading Completed at the National Museum of China to Enhance Visitor Experience",
"link":
"https://www.chnmuseum.cn/zx/gbxw/202507/t20250728_272269.shtml",
"snippet": "Humanized services were the focus of this renovation. During the renovation project, additional facilities such as toilet armrests and height-adjustable hand wash basins were added to facilitate the disabled, children, and other groups; small shelves and coat hooks were installed inside the toilet stalls to meet visitors' needs for placing personal items...",
"date": "Jul 28, 2025", "position": 1 ... |
| **Browsed Webpage** | https://www.chnmuseum.cn/zx/gbxw/202507/t20250728_272269.shtml |
| **Browsed Webpage Content** | ...In terms of functional layout, addressing the restroom distribution issues raised by the audience, this renovation involved swapping the men's and women's sections of the ground-floor restroom on the north side. While maintaining the same number of men's stalls, the number of women's stalls was increased. Additionally, the cleaning room was relocated outward, further improving space utilization efficiency.... |
| **Gold Specific Webpage** | https://www.chnmuseum.cn/cg/ |
| **Gold Specific Information** | This is the service page of the National Museum of China, which contains comprehensive visitor information. Key details include the opening hours, reservation rules, and a map displaying the exhibition halls and public facilities (e.g., restrooms). |

# F LIMITATIONS AND FUTURE WORK

This paper highlights the overlooked problem of UIS, introducing the dedicated benchmark UIS-QA and a strong baseline method, UIS-Digger. While the results presented above are promising, several limitations remain to be addressed.

First, as shown in Fig. 5, UIS-Digger continues to improve after RFT training, but the gains are limited. This suggests that despite our careful data generation and filtering pipeline, the synthesized QA distribution may still differ from real-world cases. Moreover, the sparse supervision signal—focused solely on final answers—restricts the model's ability to distinguish between trajectories that are equally correct but vary in quality.

Second, because websites evolve unpredictably, even carefully chosen time-invariant sources may shift in accessibility. For example, new third-party websites might replicate the target information, effectively transforming a UIS case into an IIS one and altering the problem difficulty.

Looking forward, we plan to enhance UIS-Digger with more advanced self-improvement techniques such as reinforcement learning, and to synthesize higher-quality QA pairs that better reflect the complexity of real-world UIS scenarios.

## G    STATEMENT ON THE USE OF AI

AI techniques were employed solely to assist with language polishing and improving sentence fluency during the writing of this paper. All ideas, methods, and experimental results were conceived, designed, and executed entirely by the authors.

## H    ETHICS STATEMENT

This work involves human annotators in the data collection process. All annotators were compensated above the minimum wage specified by the local government. The primary goal of this research is to support the community in advancing UIS-capable agents. To promote transparency and reproducibility, the dataset will be open-sourced. No commercial or confidential information is included in the dataset.

