# OpenReview forum: "UIS-Digger: Towards Comprehensive Research Agent Systems for Real-world Unindexed Information Seeking"
_ICLR.cc/2026/Conference — ICLR 2026 Poster_

### Official Review · Reviewer_oSEj · 2025-10-19

**Soundness:** 3
**Presentation:** 3
**Contribution:** 3
**Rating:** 6
**Confidence:** 4

**Summary:**

This paper introduces the problem of Unindexed Information Seeking, where vital information is inaccessible to search engines.
The authors propose UIS-QA, a benchmark of 110 expert-annotated questions requiring interactions beyond indexed snippets, and present UIS-Digger, a multi-agent system with dual-mode browsing and file reading capabilities.
Experiments show that UIS-Digger achieves 27.3% accuracy, outperforming stronger LLMs and existing multi-agent systems on UIS-QA, highlighting the importance of addressing UIS for real-world research agents.

**Strengths:**

- It identifies a novel and important gap that existing benchmarks overlook, and proposes the first so-called UIS benchmark with carefully validated data.
- Presents a multi-agent architecture with dual-mode browsing and file parsing.
- Strong empirical evaluation with comparisons against diverse baselines.
- Detailed error analysis and ablation studies provide useful insights.

**Weaknesses:**

- Absolute performance on UIS-QA remains very low (27.27%), raising concerns about practical utility.
- Training relies heavily on synthetic QA pairs, which may not fully reflect real-world UIS complexity.

**Questions:**

- How do you ensure UIS-QA’s representativeness given its small scale? Though 100+ samples are common, how do you ensure that diversity?
- How sensitive is UIS-Digger to backbone choice? Results suggest raw model quality still dominates. Does the framework generalize to weaker LLMs?

---

> ### Author Response · Authors · 2025-11-21
> **Reply to Reviewer oSEj weakness 1 and 2**
>
> Thanks a lot for your positive feedback, we value all the comments and questions, and would like to respond to them one by one in the following:
>
> # W1: "27.27% remains a low accuracy on UIS-QA test set"
>
> Although the absolute value (27.27%) of our UIS-Digger agent system on the UIS-QA test set is not impressively high, we want to stress three aspects:
>
> 1. As we also stated at the end of the *Introduction* section, the major contribution of this work is to identify and highlight a previously neglected problem of ​*unindexed information seeking*​, while UIS-Digger serves as a fairly good baseline method but not the ultimate solution. Actually, as we also mentioned in the *Conclusion* section, the UIS problem remains an open question and therefore still has its value in encouraging more attention from future works on solving UIS problems.
> 2. Even though the UIS problem has not been totally solved by our UIS-Digger, it already yields the best performance, especially with a merely 38B backbone LLM. Considering UIS-QA is quite a challenging task, even strong baselines with powerful backbone LLMs (memento+O3) can only yield at most 25.5%. The 27.27% accuracy from our UIS-Digger with a much weaker 38B backbone LLM is therefore not a trivial result.
> 3. Albeit the overall score of 27.27% of our UIS-Digger in Table 2 is still low for practical use, a positive trend can be observed from Table 4. Using a stronger model such as O3, UIS-Digger can directly achieve a higher score of 30.9. This finding suggests that with proper action space and workflow settings such as in UIS-Digger, improving the automatic ability of backbone LLMs can still improve the final results on UIS tasks.
>
> Concluding the above points, we believe that UIS-QA is a challenging and open question worthy of more future research, and UIS-Digger serves as a pioneer method showing that, with proper action space and workflow settings, small LLMs can still yield SOTA performance after tuning. On the other hand, UIS-Digger also shows an orthogonal trend with backbone LLM choices: with stronger LLMs, UIS-Digger exhibits expected improvements on the final score.
>
> # W2: "Training data are heavily synthetic, may not fully reflect real-world UIS complexity"
>
> Thanks for your concern. Indeed, the training data are largely synthetic, because of cost and time. However, the use of synthetic data for training is a common practice in the domain of information-seeking agents, e.g., WebSailor [3], WebExplorer [4], WebAggregator [5]. Moreover, in Fig. 5 we show that the synthetic training data do bring significant improvements to the agents, from 9.1% to 27.3%, suggesting that the synthetic data can serve as a good approximation to real-world (manually labelled) UIS problems.

---

> ### Author Response · Authors · 2025-11-21
> **Reply to Reviewer oSEj question 1 and 2**
>
> # Q1: "UIS-QA's diversity given the small scale"
>
> Thank you for the comment. Constructing questions with unindexed characteristics is challenging, as each instance must satisfy strict criteria. We performed rigorous filtering and excluded hundreds of candidate questions that did not fully meet the unindexed requirement.
>
> We also want to stress that benchmarks with test sets in the hundred level are common, such as Mind2Web2 [1] (130 examples) and xBench [2] (100 examples). Moreover, although UIS-QA contains a similar number of questions, each task has high complexity, and the total workload is not small despite the scale of 110 questions.
>
> As mentioned in Section 2.2, we pay special care during the dataset collection, asking the annotators to make at most 2 questions from the same website, and encouraging them to explore varieties of different domains. Please also refer to the table below: we summarize all the categories of queries in UIS-QA, justifying the diversity of UIS-QA, ranging from X domains:
>
> | Field          | Primary | Relevant |
> | ---------------- | --------- | ---------- |
> | Sports         | 5.45%   | 5.45%    |
> | Politics       | 5.45%   | 16.36%   |
> | Entertainment  | 0.91%   | 1.82%    |
> | Society        | 12.73%  | 59.09%   |
> | Science        | 7.27%   | 28.18%   |
> | Arts           | 1.82%   | 7.27%    |
> | Gaming         | 6.36%   | 6.36%    |
> | Education      | 10%     | 26.36%   |
> | IT             | 6.36%   | 10%      |
> | Nature         | 2.73%   | 7.27%    |
> | Law            | 2.73%   | 7.27%    |
> | Economics      | 24.55%  | 43.64%   |
> | Literature     | 2.73%   | 6.36%    |
> | Transportation | 1.82%   | 2.73%    |
> | Lifestyle      | 3.64%   | 10.91%   |
> | Medicine       | 5.45%   | 6.36%    |
>
> # Q2: "How sensitive is UIS-Digger to backbone? Does UIS-Digger generalize to weaker LLMs"
>
> We evaluate UIS-Digger with three extra backbone LLMs, GPT-oss-120B, O3-mini, and Qwen3-235b-a22b. Combining with the backbone ablation results in the appendix of our main paper, we list an overall table below:
>
> | backbone          | VLM        | UIS-QA Acc |
> | ------------------- | ------------ | ------------ |
> | Pangu-38B (tuned) | GPT-4o     | 27.3       |
> | Pangu-38B (tuned) | QwenVL-max | 25.5       |
> | Qwen3-32B (tuned) | GPT-4o     | 27.3       |
> | O3-mini           | GPT-4o     | 7.6        |
> | GPT-4o            | GPT-4o     | 8.2        |
> | Qwen3-235b-a22b   | GPT-4o     | 13.9       |
> | GPT-oss-120b      | GPT-4o     | 19.9       |
> | Claude-sonnet-4   | GPT-4o     | 23.6       |
> | O3                | GPT-4o     | 30.9       |
>
> From the observation in the above table, we find that the raw model quality is a precondition for the multi-agent system to perform well. Relatively old (GPT-4o) and small (O3-mini) models yield low performance on the challenging UIS-QA benchmark. Nevertheless, using the same backbone LLM of O3-mini, UIS-Digger still outperforms the comparable baseline OWL+O3-mini (4.6%).
>
> Another key finding is that once the backbone model's raw ability is not too weak, the final scores on UIS-QA align well with the model's quality. By deploying different models from Qwen3-235b-a22b to GPT-oss-120b, Claude-sonnet-4, and finally to O3, the overall accuracy consistently increases from 13.9% to 30.9%, showing a good regularity of the UIS-Digger framework. It is worth noting that using relatively weaker backbones of Qwen3 and GPT-oss, UIS-Digger can already surpass most of the baseline methods except Memento and Tongyi-DR, which deploy strong backbones of O3 or heavily tuned specific models.
>
> ---
> Reference:
>
> [1] Boyu Gou, et al. "Mind2Web 2: Evaluating Agentic Search with Agent-as-a-Judge". arXiv preprint arXiv:2506.21506
>
> [2] Kaiyuan Chen, et al. "xbench: Tracking Agents Productivity Scaling with Profession-Aligned Real-World Evaluations". arXiv preprint arXiv:2506.13651
>
> [3] Li, Kuan, et al. "WebSailor: Navigating Super-human Reasoning for Web Agent." *arXiv preprint arXiv:2507.02592* (2025).
>
> [4] Liu, Junteng, et al. "Webexplorer: Explore and evolve for training long-horizon web agents." *arXiv preprint arXiv:2509.06501* (2025).
>
> [5] Wang, Rui, et al. "Explore to Evolve: Scaling Evolved Aggregation Logic via Proactive Online Exploration for Deep Research Agents." *arXiv preprint arXiv:2510.14438* (2025).

---

### Official Review · Reviewer_9Jka · 2025-10-31

**Soundness:** 2
**Presentation:** 3
**Contribution:** 3
**Rating:** 6
**Confidence:** 3

**Summary:**

This paper tackles the underexplored challenge of Unindexed Information Seeking (UIS) — retrieving essential information unavailable through standard search engines. The authors introduce UIS-QA, a benchmark specifically designed to evaluate agent performance on UIS tasks, and propose UIS-Digger, an agent system equipped with enhanced web-interactive tools and trained via sequential supervised and reinforcement fine-tuning. Experiments show that UIS-Digger achieves state-of-the-art results on UIS-QA, though overall accuracy (27.27%) highlights the inherent difficulty of the task. The work is valuable for formalizing UIS as a benchmarkable problem and demonstrating a viable training strategy, but the gains remain modest and practical effectiveness is still limited.

**Strengths:**

1. This work identifies an interesting unindexed information-seeking problem, which is practical in reality and still under-explored.
2. This work introduces a dataset to study this problem and also constructs an agent to solve this challenge correspondingly.

**Weaknesses:**

1. The scope definition of UIS is not crystally clear. How is “unindexed” information precisely defined? Does it include API-gated, or private web data?
2. The boundary with traditional search is not super clear as well. How to distinguish UIS tasks from regular information-seeking tasks where the answer is simply poorly ranked or paraphrased online?
3. This work claims to build a benchmark, but the characteristics of the benchmark are not enough. For example, what fraction of examples involve dynamic pages, paywalls, or databases?

**Questions:**

1,  Is there any overlap between UIS-QA examples and the pretraining or fine-tuning corpora (e.g., cached webpages or documentation dumps)? Can using DeepSeek to filter out data address this?
2. Can UIS-Digger transfer to new domains or unseen data environments, or is it overfit to UIS-QA’s structure?

---

> ### Author Response · Authors · 2025-11-21
> **Reply to Reviewer 9Jka  weakness 1 and 2**
>
> # Response to Reviewer 9Jka
>
> We are grateful to the reviewer for the positive review and detailed suggestions, which have contributed to strengthening our manuscript. Below, we provide our responses to each comment.
>
> ## W1. Definition and Scope of UIS
>
> Please also refer to Section 2.1 for a more detailed definition of UIS. Strictly speaking, any information accessible from the internet, but cannot be retrieved by search engine is defined as as unindexed information.
>
> However, the information behind private web or privileged APIs are not in the scope of this research, because it's no longer a technical problem but rather a property problem. For the same reason, violently trying all possible search queries and navigate all the returned pages is also not in the scope of this research because it is simply infeasible and have no practical meaning.
>
> In this paper, we only discuss unindexed information that are publicly accessible, but cannot be directly returned by search engines in reasonable effort, as stated in Eq. 4 to 6. As also mentioned in the manuscript, the UIS-QA mainly presents a reasonable approximation of the whole potential unindexed information hiding in the vast internet. **Please also refer to Table 3 in our reply to Weakness 3, for typical types of UIS problems.**
>
> Nevertheless, our experimental results show that agent systems such as UIS-Digger designed and aware of UIS problem can still achieve enhanced capabilities in UIS problems. Even using a small backbone, it demonstrate good generality towards real-world human labelled UIS questions.
>
> ## W2. Distinction between UIS and traditional IIS
>
> A good distinction is to check whether the information can be finally retrieved by a search engine by providing more precise and comprehensive search queries. Using this method, it is obvious that the two scenarios proposed in your comments, when the target information is simply poorly ranked, should fall within the scope of traditional IIS. Because in this case, a smarter searching strategy or a better search query, could still theoretically retrieve target information directly from the search engine.
>
> However, the UIS problems in UIS-QA, we apply manually search using the exact original webpage title as query, but the search engine still cannot directly retrieve the target information. This suggests a clear distinction between 'difficult IIS' problems and UIS problem, whether the search engine is totally unaware of the target information, even if golden search queries are provided.
>
> Reasons that cause a piece of information being 'unindexed information' are various, we include them in our reply to weakness 3 (table 3).

---

> > ### Author Response · Authors · 2025-11-21
> > **Reply to Reviewer 9Jka weakness 3**
> >
> > ## W3. The characteristic distribution of UIS-QA
> >
> > Thank you for your suggestion, we enhance the statistics of our UIS-QA problem. More details are provided below in three aspects: language, knowledge domain, and UIS type. Corresponding statistical summaries are provided in below Table 1, Table 2, and Table 3, respectively. We offer further clarification on certain details within these tables as follows.
> >
> > **1. Language**:
> >
> > > UIS-QA is a bilingual benchmark, consisting of 76% Chinese and 24% English questions. Notably, 9% of the Chinese questions also contain English terms or phrases. This design necessitates that agent systems be capable of searching across both Chinese and English websites, underscoring the linguistic balance and comprehensiveness of the benchmark.
> >
> > **2. Knowledge Domains**:
> >
> > > To ensure that each UIS question can thoroughly and comprehensively evaluate the practicality and generalization ability of Agent frameworks, every question in UIS-QA has been meticulously selected and annotated from a large pool of candidate questions. As a result, each question inherently spans multiple knowledge domains. For each question, we have identified the primary knowledge domain of the target information (the *Dominant* column), as well as the top three knowledge domains relevant to both the question content and potential search trajectory (the *Relevant* column).
> > > Statistically, the target information referenced in the 110 questions of UIS-QA is relatively evenly distributed across 16 domains (as shown in the *Dominant* column of Table 2). Furthermore, each knowledge domain is also covered by at least more than one question (as reflected in the *Relevant* column of Table 2).
> >
> > **3. UIS Types**:
> >
> > > From a high-level perspective, we have categorized UIS questions into four types based on the medium in which the target information resides and the potential search paths required:
> > >
> > > - Deep Navigation: the target is deeply hidden in sub-level pages of a website (too deep that not indexed by search engine or shown on main pages)
> > > - Ajax/Widget Interaction: the target is hidden in backend database and requires complex operations or dynamically interactive components
> > > - File Reading: the target is hidden in files, including PDF, xlsx etc., which are not indexed and can not be directly returned from search engines
> > > - Multimodal Perception: the target is contained in multimodal data structure, including image, dynamic charts etc., which can not be directly retrieved from search engine returned snippets.
> > >
> > > Similar to the **knowledge domain** dimension, to ensure that UIS-QA provides a comprehensive evaluation of Agents' capabilities, 82.7% (91 out of 110) of the questions involve two or more UIS types, which means Agents must overcome multiple distinct UIS challenges to obtain the final answer.
> > > Among these, Deep Navigation, as a relatively foundational capability for solving UIS tasks, is present in nearly all questions (93.6% coverage).
> >
> > Based on the above statistical evidence, we demonstrate that UIS-QA not only has pioneered a new research topic in information seeking but also constitutes a comprehensive, in-depth, and challenging benchmark, providing substantial support for the advancement of the research community.
> >
> > **Table 1: Language Distribution of UIS-QA Benchmark**
> >
> > |Language|Chinese|English|
> > |-|-|-|
> > |Proportion (Quantity)|76.4% (84)|23.6% (36)|
> >
> > **Table 2: Field Distribution of UIS-QA Benchmark**
> >
> > |Field|Dominant|Relevant|
> > |-|-|-|
> > | Sports | 5.45% (6) | 5.45% (6) |
> > | Politics | 5.45% (6) | 16.36% (18) |
> > | Entertainment | 0.91% (1) | 1.82% (2) |
> > | Society | 12.73% (14) | 59.09% (65) |
> > | Science | 7.27% (8) | 28.18% (31) |
> > | Arts | 1.82% (2) | 7.27% (8) |
> > | Gaming | 6.36% (7) | 6.36% (7) |
> > | Education | 10.00% (11) | 26.36% (29) |
> > | IT | 6.36% (7) | 10.00% (11) |
> > | Nature | 2.73% (3) | 7.27% (8) |
> > | Law | 2.73% (3) | 7.27% (8) |
> > | Economics | 24.55% (27) | 43.64% (48) |
> > | Literature | 2.73% (3) | 6.36% (7) |
> > | Transportation | 1.82% (2) | 2.73% (3) |
> > | Lifestyle | 3.64% (4) | 10.91% (12) |
> > | Medicine | 5.45% (6) | 6.36% (7) |
> >
> > **Table 3: UIS Type Distribution of UIS-QA Benchmark**
> >
> > |UIS Type|Dominant|Relevant|
> > |-|-|-|
> > | File Parsing | 25.45% (28) | 28.18% (31) |
> > | Ajax/Widget Interaction | 32.73% (36) | 51.82% (57) |
> > | Deep Navigation | 24.55% (27) | 93.64% (103) |
> > | Multimodal Perception | 17.27% (19) | 27.27% (30) |

---

> > > ### Author Response · Authors · 2025-11-21
> > > **Reply to Reviewer 9Jka question 1 and 2**
> > >
> > > ## Q1. overlap between UIS-QA examples and the corpora, can Deepseek filtering address this?
> > >
> > > The overlap is minimal. We do not have access to the pre-training or post-training corpora for most open- and closed-source LLMs. As a result, we design an experiment where the model relies solely on its parameter memory for reasoning (without accessing any search tools or external knowledge) to probe the degree of overlap. The results, as shown in Table 2 of the main paper, indicate a very low degree of overlap, with Claude-Sonnet 4 answering only 2.7% of the questions correctly and GPT-5 answering only 0.9%. These results demonstrate that the effective memory of the models is minimal, and they have not retained much of the information. Such a small overlap does not significantly impact the validity or resolution of the evaluation.
> > >
> > > Additionally, using DeepSeek-R1’s internal knowledge, without relying on external tools, for filtering is based on the same principle. The above 2.7% and 0.9% minimal overlap is exactly achieved by using DeepSeek as the filter. The results demonstrate that DeepSeek as a filter can yields highly-qualified outcomes.
> > >
> > > Furthermore, all questions in the UIS-QA benchmark are categorized as Unindexed Information, which have been carefully screened and verified. These questions are intuitively difficult to capture via conventional pre-training data crawling processes. Therefore, sparse overlap between UIS-QA and training/fine-tuning corpora is an expected outcome.
> > >
> > > As a result, the questions in our UIS-QA benchmark are novel, challenging, and resistant to simple memorization, thus effectively assessing agents' comprehensive ability to seek information on the internet.
> > >
> > > ## Q2. Generalization capability of UIS-Digger
> > >
> > > UIS-Digger is designed to be general, and not ad-hoc tailored for the UIS-QA benchmark, exhibiting strong generalization capability. The multi-agent system design demonstrate generic atomic abilities of LLMs, and therefore can generalize to other tasks.
> > >
> > > To evaluate the generalization ability of our framework and training methods, we tested UIS-Digger on three additional datasets beyond UIS-QA: Browsecomp-zh, GAIA, and FinSearchComp.
> > >
> > > - Browsecomp-zh: A challenging dataset involving **long, multi-step fuzzy retrieval** questions. It assesses the model's ability to use search engines over multiple rounds, design search queries, manage ultra-long contextual chains, and perform information discrimination.
> > > - GAIA: A multi-tool evaluation benchmark that tests the model's **ability to use multiple tools** effectively to complete tasks.
> > > - FinSearchComp: A dataset annotated by financial professionals based on specialized financial websites. This dataset shows the UIS-Digger's ability in vertical domains, where the model needs to **generalize to unseen websites and professional questions**, such as financial websites of Bloomberg and Refinitiv.
> > >
> > > In both the SFT and RFT phases, UIS-Digger showed consistent improvement on the generalization evaluation sets: even though our framework and training data were specifically designed for UIS-QA, after conducting SFT and RFT training, the model demonstrated stable performance improvements on Browsecomp-zh, GAIA, and FinSearchComp. In the SFT phase, UIS-Digger improved by 18.7% on Browsecomp-zh, 17.5% on GAIA, and 11.6% on FinSearchComp. In the RFT phase, UIS-Digger further improved by 1.7% on Browsecomp-zh and 7.8% on GAIA, demonstrating the excellent generalization capability of our training data and methodology.
> > >
> > > The overall score of UIS-Digger is 42.2, surpassing GPT-5 Thinking (39.5) and 1T-parameter Kimi-k2 (39.9). Our method ranks No. 3, according to the official FinSearchComp results. These outcomes further substantiate the strong generalization capability of UIS-Digger.
> > >
> > > | |UIS-QA | BC-zh | GAIA-textual-103 | FinSearchComp(T2/T3) |
> > > |------|------|-----:|-----:|-----:|
> > > | Pangu-ZeroShot | 9.1 |12.1    | 25.2   | 25.7(48.0/3.4) |
> > > | Pangu-SFT | 22.7| 30.8   | 42.7   | 37.3(69.0/5.7)   |
> > > | Pangu-RFT | 27.3| 32.5    | 50.5   | 42.2(73.0/11.4)   |

---

### Official Review · Reviewer_xVJL · 2025-10-31

**Soundness:** 2
**Presentation:** 3
**Contribution:** 2
**Rating:** 2
**Confidence:** 4

**Summary:**

This paper introduces the concept of "Unindexed Information Seeking" (UIS), which refers to information that cannot be directly retrieved through search engine queries and requires deeper interaction with websites (e.g., clicking, form filling, file downloading). The authors formalize the distinction between indexed information (II) - content available in search snippets or one-step crawling - and unindexed information (UI) - everything else requiring interactive navigation. They contribute: (1) UIS-QA, a benchmark of 110 expert-annotated QA pairs designed to test UIS capabilities, (2) empirical evidence showing state-of-the-art agents experience 47-54% performance drops on UIS-QA compared to existing benchmarks like GAIA, and (3) UIS-Digger, a multi-agent system with dual-mode browsing trained via SFT and RFT that achieves 27.27% accuracy, outperforming baselines including those using o3 and GPT-4.

**Strengths:**

1. **Valuable dataset contribution with clear curation criteria**: The manually created dataset of 110 validated QA pairs that explicitly avoid search engine shortcuts represents a concrete contribution.
2. **Valuable empirical evaluation**: Testing 13+ baseline systems across multiple categories provides valuable comparisons.
3. **Detailed failure analysis**: The breakdown of error modes and tool usage evolution across training stages offers useful insights.

**Weaknesses:**

1. **Insufficient differentiation from existing web agent research and benchmarks**: The paper claims that UIS represents an "underexplored challenge" and a "critical blind spot" in current agent systems, yet the capabilities required (e.g., interactive navigation, form filling, file downloading, multi-step exploration) are precisely what existing web agents can already do. The action space that UIS-Digger employs (search, crawl, click, scroll, type, download files) already exists in prior systems. Benchmarks like WebArena and Mind2Web also evaluate complex web interactions across diverse websites. The distinction appears to be one of benchmark curation methodology (filtering out search-solvable questions) rather than a discovery of a new research problem requiring novel capabilities.
2. **Critical missing baselines**: The experimental evaluation contains glaring omissions that prevent assessment of whether UIS is actually an unsolved problem for current systems. Most notably, the paper cites OpenAI Deep Research, Google Gemini Deep Research, and Grok-3 Deep Research in the related work but does not evaluate any of these systems. Recent web agent systems are absent. Without evaluating actual state-of-the-art deep research systems and web agent systems, the paper cannot substantiate its central claim that UIS represents a fundamental limitation or that the proposed 27.27% score represents competitive performance.
3. **Limited technical novelty**: UIS-Digger employs standard components from existing work: multi-agent architecture with specialized sub-agents; dual-mode browsing switching between textual and visual observation; the tools (search, crawl, browser automation, file readers); and the SFT + RFT training pipeline is a standard approach. More problematically, the system achieves only 27.27% accuracy, barely outperforming the Memento baseline at 25.5% (which was tested in a handicapped configuration without its case bank). This marginal improvement despite task-specific training on synthesized data similar to the test distribution might suggest that the paper has not discovered an effective solution methodology.

**Questions:**

1. WebArena and Mind2Web already test interactive navigation. What would be the difference between UIS-QA and them?
2. What would be the performance of UIS-Digger with the SoTA LLMs (like GPT, Claude) as the backbone?

---

> ### Author Response · Authors · 2025-11-21
> **Reply to Reviewer xVJL weakness 1 (part1)**
>
> We highly appreciate your efforts and comments, and we would like to address your concerns one by one.
>
> # W1: Insufficient differentiation from existing web agent research and benchmarks
>
> In the following table, we analyze the major differences between our UIS-QA benchmark and existing datasets along five dimensions. We hope that this clarification makes it clear that UIS-QA–style **open-domain information-seeking tasks** are fundamentally different from Computer/Browser Use tasks. The latter begin from an explicitly specified starting website and therefore do not require any exploration within the open web environment, while UIS-QA evaluates the organic combination of information-seeking capabilities and computer/browser use capabilities of agent systems.
>
> |                        | Type         | Real-World Web Environment | Unknown Startpoint | Unindexed-Information Dependence | Final Answer-oriented Evaluation |
> | ------------------------ | -------------- | ---------------------------- | -------------------- | ---------------------------------- | ---------------------------------- |
> | WebArena[1]            | Computer Use | ×                         | ×                 | -                                | ×                               |
> | Mind2Web[2]            | Computer Use | ×                         | ×                 | -                                | ×                               |
> | Mind2Web-Live[3]       | Computer Use | √                         | ×                 | √                               | ×                               |
> | Online-Mind2Web[4]     | Computer Use | √                         | ×                 | √                               | ×                               |
> | Browsecomp-en/zh[5][6] | Info Seeking | √                         | √                 | ×                               | √                               |
> | xbench-DeepSearch[7]   | Info Seeking | √                         | √                 | ×                               | √                               |
> | GAIA-textual-103[8][9] | Info Seeking | √                         | √                 | ×                               | √                               |
> | **UIS-QA**       | Info Seeking | √                         | √                 | √                               | √                               |
>
> ## 1. Task Type
>
> Recent works on agent capabilities and benchmarks have largely evolved along two relatively independent trajectories: Info-seeking and Computer Use. They focus on very different aspects of agent capabilities. Our ​**UIS-QA is a unique information-seeking task that also requires capabilities from computer use**​.
>
> ### Information-seeking Tasks
>
> ---
>
> These tasks require the agent to conduct multi-step exploration on the open web to acquire information. The main challenges include: search query formulation and refinement, discrimination of conflicting information, analysis and reasoning over long texts, and multi-step reflection and self-correction. This kind of task focuses on ​**search strategy & information extraction**​.
>
> * **Benchmarks:** HotpotQA, SimpleQA, FRAMES, Browsecomp-en/zh, GAIA, xbench-DeepSearch.
> * **Agents:** Typical proprietary systems such as OpenAI DeepResearch, Google Gemini Deep Research, and Grok-3 Deep Research; and open-source systems such as Tongyi DeepResearch, DeepResearcher, and Search-r1.
> * **Typical query [5][6]:**
>   *Between 1990 and 1994 (inclusive), what teams played in a soccer match with a Brazilian referee that had four yellow cards, two for each team, where three of the total four were not issued during the first half, and four substitutions, one of which was for an injury in the first 25 minutes of the match?*
>
> ### Computer Use Tasks
>
> ---
>
> These tasks focus on operating a computer or browser to complete user-specified goals, through interactive operations (type/click/scroll/hover/swipe). This kind of task focuses on ​**operating software**​.
>
> * **Benchmarks:** WebArena, Mind2Web, Online-Mind2Web, Mind2Web-Live, Mind2Web 2
> * **Agents:** Proprietary systems such as Claude Computer Use and OpenAI Operator; and open-source systems such as SeeAct, Browser Use, and Agent-E.
> * **Typical query [1][3]:**
>   *Post a question: whether I need a car in NYC (on Reddit).*
>   *Locate the list of movies “at home”, sorted by most recent in rottentomatoes.*
>
> ## 2. Real-World Web Environment
>
> Unlike WebArena and Mind2Web, UIS-QA evaluates agents in a real-world web environment, where agents interact directly with the live, public Internet rather than with a locally hosted or simulated website collection. In a real-world web environment, agents must have capabilities in handling genuine web noise such as conflicting or outdated information, secondary sources, distracting content, complex page structures, pop-ups, and advertisements. On the other hand, in self-hosted or simulated websites, agents seldom encounter these challenges.

---

> ### Author Response · Authors · 2025-11-21
> **Reply to Reviewer xVJL weakness 1 continue (part2)**
>
> ## 3. Unknown Startpoint
>
> Moreover, UIS-QA, as an information-seeking task, has no startpoint provided; therefore the agents need to search across the *entire* public web via general-purpose search engines (e.g., Google), without a narrowed-down scope of sites. This more realistic task setting poses higher requirements on agents in multi-step exploratory searching, following links across domains, and reflecting their strategies. Hence UIS-QA not only tests an agent's ability in deciding **how** to navigate a single given site, but also **where** to seek information from the open web.
>
> In most Computer Use benchmarks (WebArena, Mind2Web, Online-Mind2Web [1][2][3][4]), the target website (e.g., Reddit, Yelp, Uniqlo, ESPN, etc.) is explicitly specified in the user query (e.g., Find a Ford Mustang with lowest price and save it in kbb). As a result, the agents do *not* need to perform web-scale exploration via a search engine: the search space is essentially constrained to a known site, which sidesteps many of the core challenges of open-domain information seeking.
>
> By contrast, UIS-QA is designed explicitly around ​**open-web search and exploration**​: the agent must first decide which sites to visit and then navigate them effectively.
>
> ## 4. Unindexed-Information Dependence
>
> As mentioned in the manuscript, UIS-QA differs from all existing information-seeking benchmarks by explicitly introducing ​**Unindexed-Information Dependence**​, where pivot information for solving the task is *not* directly reachable via standard search engine results.
>
> In previous Info-seeking benchmarks [5][6][7][8], this dimension has been largely overlooked. In practice, many datasets can be solved to a high degree of accuracy using only search + crawl pipelines, as demonstrated by systems such as Tongyi DeepResearch, kimi k2 thinking, and glm4.6. The scores on these benchmarks mainly reflect indexed information-seeking ability but cannot effectively represent **unindexed information-seeking** capabilities [5][6][7][8][9].
>
> In contrast, UIS-QA is explicitly constructed to have high Unindexed-Information Dependence. Building on previous methodologies [4], we employ a stronger model and larger computation budget: we use o3 together with search tools for multiple rounds of automatic filtering, followed by careful human screening, to ensure that no indexed-information shortcut exists. This is precisely the aspect that distinguishes UIS-QA from existing Info-seeking benchmarks, beyond a mere “curation style” difference.
>
> ## 5. Final-answer-oriented Evaluation
>
> The UIS-QA benchmark adopts deterministic short-form answers as the criterion for judging agents' performance, which is straightforward and robust. In contrast, computer use tasks such as WebArena or Mind2Web usually rely on inspecting intermediate actions (e.g., whether the agent correctly clicks a button), making the evaluation process sophisticated and less aligned with human intuition. By contrast, UIS-QA adopts final-answer-oriented evaluation and consequently **significantly reduces evaluation noise and allows automatic evaluation**.

---

> ### Author Response · Authors · 2025-11-21
> **Reply to Reviewer xVJL weakness 2**
>
> # W2: Critical missing baselines
>
> We add new baseline results of OpenAI Deep Research. Combining the results from the main table and appendix in our paper, we have the following conclusions:
>
> **UIS problem is highly under-explored:**
> As already reported in both the main results table and the appendix, we evaluate the performance of OpenAI o3 and Claude-Sonnet-4. Even with such strong backbones, they yields at most 30.9% and 23.6%, respectively. Additionally, OpenAI Deep Research is also tested, and yields merely 21.7% accuracy on UIS-QA. We also asked 6 human testees to work on UIS-QA, and their average score is 48.2%, still substantially higher than these advanced agent systems, further underscoring that UIS-QA is far from being a solved problem. By contrast, the current best systems already reach 60.2% on Browsecomp [10], and the highest reported score on GAIA is 87.04%. From both a relative and an absolute perspective, UIS-QA evidently remains a novel and unsolved problem.
>
> **UIS-Digger yields the highest score with a much smaller backbone LLM**
> Even though the UIS problem has not been totally solved by our UIS-Digger, it already yields the best performance among all the baselines, especially with a merely 38B backbone LLM. Considering UIS-QA is a quite challenging task that even strong baselines with powerful backbone (e.g., memento+O3) can only yield at most 25.5%, the 27.27% accuracy from our UIS-Digger achieved with a much weaker 38B backbone LLM is not a trivial result.
>
> **UIS-Digger is an effective framework and even surpass close-sourced frameworks like Deep Research**
> Comparing the second and fifth rows of the table, we observe that o3, when equipped with the UIS-Digger framework, improves from 25.5% to 30.9%, which demonstrates that the UIS-Digger framework provides a substantial benefit for solving UIS-QA.
>
> |   | Main Model        | Scaffold                                | UIS-QA Score |
> | --- | ----------------- | ----------------------------------------- | -------------- |
> | 1 | Claude4-Sonnet  | UIS-Digger Framework                    | 23.6         |
> | 2 | o3              | Memento                                 | 25.5         |
> | 3 | Pangu-38B(ours) | UIS-Digger Framework                    | 27.3         |
> | 4 | Qwen3-32B(ours) | UIS-Digger Framework                    | 27.3         |
> | 5 | o3              | UIS-Digger Framework                    | 30.9         |
> | 6 | o3             | OpenAI Deep Research(proprietary system) | 20.0|

---

> ### Author Response · Authors · 2025-11-21
> **Reply to Reviewer xVJL weakness 3 and question 1**
>
> # W3: Technical Novelty
>
> As we also stated at the end of the 'Introduction' section, the major contribution of this work is to identify and highlight a previously neglected problem of 'unindexed information seeking', while the UIS-Digger serves as a fairly good baseline method but not the ultimate solution. Even though, this work still exhibits technical novelty in problem formulation, data synthesis and workflow design.
>
> Although there are existing works including textual and multimodal browser, they mainly takes them as two separate tools but not an organic whole. To the best of our knowledge, none of them have enabled a single agent to flexibly and seamlessly switch between two different backbone models (LLM for stronger reasoning and VLM for visual ability). In our design, the agent can actively decide which one to invoke. Moreover, we enable the agent to share task execution history (i.e., consistent memory) seamlessly during using the different backbone models. In UIS-Digger, the agent maintains a shared task execution history (i.e., a consistent memory) across both backbones, enabling consistent memory of task performing. Furthermore, the text and graphical modes of our browser operate on the same underlying webpage state, so the agent does not need to manually reconstruct or synchronize the navigation context when switching modes. This dual-mode design greatly reduces the cost of context management and significantly improves task execution efficiency.
>
> As also noted earlier, the Info-seeking community lacks training data tailored to UIS tasks. We are the first to innovatively design a data synthesis framework capable of generating such training data. In particular, through environment simulation, we construct training data that specifically reinforces designated actions (see Section 3.2.1). This dataset is of high quality and effectively boost our \~30B-parameter models to reach performance comparable to o3 (for example, after SFT, Qwen3-32B improves from 9.1% to 22.7% on UIS-QA). Please refer to Fig.5 in appendix.
>
> ---
>
> **For the question on accuracy**:
> Even though the UIS problem has not been totally solved by our UIS-Digger, it already yields the best performance, especially with a merely 38B backbone LLM. Considering UIS-QA is a quite challenging task that even strong baselines (memento) with powerful backbone LLM (O3) can only yield at most 25.5%, the 27.27% accuracy from our UIS-Digger with a much weaker 38B backbone LLM is not a trivial result. Moreover, memento's memory bank requires hundreds of historical execution records to augment the agent's experience in specific domain, which is even larger than the whole UIS-QA dataset, making it costly and unrealistic for our case.
>
> It is also important that UIS-QA dataset is quite challenging that common human testees can only achieve 48% accuracy, which further supports that the 27.27% result of UIS-Digger is not a trivial achievement.
>
> Moreover, the UIS-Digger system generalizes effectively to new domains and data environments. For example, when directly transferred to the FinSearchComp Greater China setting without any task-specific adaptation, it achieves 42.18 (73.0 @ T2), 11.4 @ T3), even outperforming GPT-5 Thinking (39.5) and the 1T-parameter Kimi-k2 (39,9). Our methods ranks at No.3, according to the official FinSearchComp results.
>
> # Q1: "difference from WebArena and Mind2Web"
>
> Please refer to our reply for Weakness 1. UIS-QA is more than an interactive navigation task.

---

> ### Author Response · Authors · 2025-11-21
> **Reply to Reviewer xVJL question2**
>
> # Q2: "performance of UIS-Digger with the SoTA LLMs (like GPT, Claude)"
>
> | backbone          | VLM        | UIS-QA Acc |
> | ------------------- | ------------ | ------------ |
> | Pangu-38B (tuned) | GPT-4o     | 27.3       |
> | Pangu-38B (tuned) | QwenVL-max | 25.5       |
> | Qwen3-32B (tuned) | GPT-4o     | 27.3       |
> | O3-mini           | GPT-4o     | 7.6        |
> | GPT-4o            | GPT-4o     | 8.2        |
> | Qwen3-235b-a22b   | GPT-4o     | 13.9       |
> | GPT-oss-120b      | GPT-4o     | 19.9       |
> | Claude-sonnet-4   | GPT-4o     | 23.6       |
> | O3                | GPT-4o     | 30.9       |
>
> Please look at Table 4 in the appendix of our paper, and also our reply to Weakness 2. We have evaluated UIS-Digger with various backbone models including GPT and Claude.
>
> UIS-Digger serves as a pioneer method showing that with proper action space and workflow settings, small LLMs can still yields SOTA performance on UIS tasks after tunning. On the other hand, the UIS-Digger also shows a orthogonal trend with backbone LLM choices, with stronger LLMs, the UIS-Digger shows expectable improvements on the final score.
>
> ---
>
> Reference:
>
> [1] Zhou, Shuyan et al. “WebArena: A Realistic Web Environment for Building Autonomous Agents.”NeurIPS 2023
>
> [2] Deng, Xiang et al. “Mind2Web: Towards a Generalist Agent for the Web.”NeurIPS 2023
>
> [3] Pan, Yichen et al. “WebCanvas: Benchmarking Web Agents in Online Environments.” *ArXiv* abs/2406.12373 (2024): n. pag.
>
> [4] Xue, Tianci et al. “An Illusion of Progress? Assessing the Current State of Web Agents.” *COLM 2025*
>
> [5] Wei, Jason et al. “BrowseComp: A Simple Yet Challenging Benchmark for Browsing Agents.”*ArXiv* abs/2504.12516 (2025): n. pag.
>
> [6] Zhou, Peilin et al. “BrowseComp-ZH: Benchmarking Web Browsing Ability of Large Language Models in Chinese.” *ArXiv* abs/2504.19314 (2025): n. pag.
>
> [7] Chen, Kaiyuan et al. “xbench: Tracking Agents Productivity Scaling with Profession-Aligned Real-World Evaluations.”*ArXiv* abs/2506.13651 (2025): n. pag.
>
> [8] Mialon, Grégoire et al. “GAIA: a benchmark for General AI Assistants.” *ArXiv* abs/2311.12983 (2023): n. pag.
>
> [9] Li, Xiaoxi et al. “WebThinker: Empowering Large Reasoning Models with Deep Research Capability.” *ArXiv* abs/2504.21776 (2025): n. pag.
>
> [10] Team, Kimi, et al. "Introducing Kimi K2 Thinking" https://moonshotai.github.io/Kimi-K2/thinking.html

---

### Official Review · Reviewer_RP5P · 2025-11-03

**Soundness:** 3
**Presentation:** 2
**Contribution:** 3
**Rating:** 6
**Confidence:** 3

**Summary:**

The paper tackles the interesting task for Unindexed Information Seeking (UIS), which involves answering questions based on vital information that is not directly surfaced by the search engines. The paper addresses this with a new UIS-QA benchmark along with the UIS-Digger approach that has a dual-browsing mode for interactively accessing information. The authors further finetune multiple backbone models using synthetic trajectories and demonstrate that the trained models outperform existing approaches on this task.

**Strengths:**

1) The experiments results are impressive, with exploration of the SFT and RFT finetuning approaches
2) The paper has a comprehensive analysis of the different actions and tool calls used.

**Weaknesses:**

1) The proposed UIS-QA benchmark is a bit limited in size with only 110 examples (with a split of 84 questions in Chinese and 26 questions in English). The authors should consider expanding the dataset to ~300 instances. Moreover, it would be worthwhile to also report expert human performance on this dataset to give a sense of upperbound.

2) While the proposed UIS-Digger models excel on the UIS-QA dataset, the performance is low on GAIA and BrowseComp-zh datasets, bringing into question the generality of the approach. The authors should also show performance of zero-shot, SFT and RFT variants on the GAIA and BrowseComp-zh datasets to give a sense of whether the finetuning degrades performance on other benchmarks.

3) The paper is missing a lot of important experimental details, particularly in how the SFT and RFT finetuning was done. Moreover, I have concerns about the reproducibility of Section 3.2.1 and Section 3.3 as minimal details have been provided.

**Questions:**

1) For the SFT and RFT processes, are the trajectories for the Planner, WebSearcher, WebSurfer and FileReader agents separately aggregated and then used for training?

---

> ### Author Response · Authors · 2025-11-21
> **Reply to Reviewer RP5P**
>
> We thank the reviewer for the valuable suggestions and insights. Below we provide our point-by-point responses.
>
> **W1: The proposed UIS-QA benchmark is a bit limited in size with only 110 examples (with a split of 84 questions in Chinese and 26 questions in English). The authors should consider expanding the dataset to \~300 instances. Moreover, it would be worthwhile to also report expert human performance on this dataset to give a sense of upperbound.**
>
> Constructing questions with unindexed characteristics is challenging, as each instance must satisfy strict criteria. We performed rigorous filtering and excluded hundreds of candidate questions that did not fully meet the unindexed requirement. Specifically, we used o3 with search tools under the ReAct framework to test all candidate questions, and those that could be answered using only indexed information were discarded. We also apply cross-validation and iterative refinement to ensure data quality. We note that several popular benchmarks, such as Mind2Web2 [1] (130 examples) and xbench-deepsearch [2] (100 examples), are of comparable scale.
>
> To provide a sense of the upper bound, we evaluated our dataset with six human experts. On the 110 samples, the overall accuaracy is 48.2%. As evaluators are allowed to skip questions after one hour of effort, approximately 20% of questions are skipped. Among the attempted questions, the average accuracy was 60.2%, with an average of 2.95 search turns, 5.4 pages browsed, 0.4 files read, and an average time of 16.5 minutes per question.
>
> **W2: While the proposed UIS-Digger models excel on the UIS-QA dataset, the performance is low on GAIA and BrowseComp-zh datasets, bringing into question the generality of the approach. The authors should also show performance of zero-shot, SFT and RFT variants on the GAIA and BrowseComp-zh datasets to give a sense of whether the finetuning degrades performance on other benchmarks.**
>
> We have conducted additional experiments on both GAIA and BrowseComp-zh, with the results now included in Appendix D.1.
>
> As shown in the table below, both SFT and RFT lead to notable performance improvements on these benchmarks, demonstrating that our approach generalizes effectively and does not degrade performance on other tasks. We have also evaluated UIS-Digger on FinSearchComp[3], a financial-domain agent benchmark, where it achieves competitive results: 73.0 on T2 and 11.4 on T3. The overall score of UIS-Digger is 42.2, even outperforming GPT-5 Thinking (39.5) and the 1T-parameter Kimi-k2 (39,9). Our methods ranks at No.3, according to the official FinSearchComp results. These outcomes further substantiate the strong generalization capability of UIS-Digger.
>
> | |UIS-QA | BC-zh | GAIA-textual-103 | FinSearchComp(T2/T3) |
> |------|------|-----:|-----:|-----:|
> | Pangu-ZeroShot | 9.1 |12.1    | 25.2   | 25.7(48.0/3.4) |
> | Pangu-SFT | 22.7| 30.8   | 42.7   | 37.3(69.0/5.7)   |
> | Pangu-RFT | 27.3| 32.5    | 50.5   | 42.2(73.0/11.4)   |
>
> **W3: The paper is missing a lot of important experimental details, particularly in how the SFT and RFT finetuning was done. Moreover, I have concerns about the reproducibility of Section 3.2.1 and Section 3.3 as minimal details have been provided.**
>
> We have expanded Section 3.2.1 to provide a comprehensive description of our QA construction pipeline, which is now also illustrated with a dedicated figure. The section now explicitly details the process for both real-world websites (including homepage selection, deep navigation, and LLM-based question generation) and simulated environments (designed to tackle specific interactive elements). To ensure reproducibility, prompts are now provided in Appendix B. Training implementation details for SFT and RFT are now included in Appendix C.
>
>
> **Q1: For the SFT and RFT processes, are the trajectories for the Planner, WebSearcher, WebSurfer and FileReader agents separately aggregated and then used for training?**
>
> Yes. The trajectories from all four agent types, Planner, WebSearcher, WebSurfer, and FileReader, are aggregated and used together to train a single backbone model.
>
> ---
> References:
>
> [1] Boyu Gou, et al. "Mind2Web 2: Evaluating Agentic Search with Agent-as-a-Judge". arXiv preprint arXiv:2506.21506
>
> [2] Kaiyuan Chen, et al. "xbench: Tracking Agents Productivity Scaling with Profession-Aligned Real-World Evaluations". arXiv preprint arXiv:2506.13651
>
> [3] Liang Hu, et al. "FinSearchComp: Towards a Realistic, Expert-Level Evaluation of Financial Search and Reasoning". arXiv preprint arXiv:2509.13160

---

### Author Response · Authors · 2025-12-01
**Letter to the AC**

Dear AC,

We noticed the recent OpenReview bug and the subsequent changes in the conference policy. Unfortunately, despite our detailed replies to all reviewer comments, the system accidentally closed the discussion, so we no longer have the opportunity to receive further feedback. In this situation, we sincerely hope you could take our replies into account and reconsider the rating of our manuscript.

To provide a clearer overview, we briefly summarize our responses:

1. Regarding concerns about the scale of UIS-QA dataset. Actually, we adopt a **quality-first, not quantity-maximizing strategy**: to ensure both diversity and reliability, we invested approximately 779 hours of human annotation and filtering. This scale is comparable to existing benchmarks such as Mind2Web2 and XBench. Moreover, the relative complexity of annotating UIS questions is much higher, due to the non-trivial effort required to filter out questions that can be directly answered by search engines using various search queries. In addition, different baseline models still perform very differently on UIS-QA (from 0.9% to 30.9%), showing a good discriminative power. As a pioneering work that first identifies the unindexed information-seeking problem, we believe the scale of UIS-QA is not a weakness.

2. Another concern is about statistics of UIS-QA. In our replies to Reviewers **oSEj** and ​**9Jka**​, we added more results and included a very detailed table. Comprehensive analyses have been conducted, showing the diversity of UIS-QA in terms of knowledge domains and UIS challenge types. We believe these replies have addressed the reviewers' concerns and demonstrated that UIS-QA has high diversity and covers the major types of common reasons that makes a piece of information 'unindexed'.

3. Some reviewers also asked for more baseline models and benchmarks for comparison. In our reply to ​**RP5P**​, UIS-Digger is tested on three non-UIS domains: BC-zh, GAIA-textual-103, and FinSearchComp (T2/T3), which represent different aspects of evaluation in heavy information digging, broad tool using, and financial domain–specific research. The results show that UIS-Digger achieves strong performance on all these datasets and continues to improve after SFT and RFT tuning using UIS training data. This suggests good generality of our approach.

4. In our replies to Reviewers **xVJL** and ​**oSEj**​, we also emphasized that we already included different backbone LLMs in the ablation study in the original paper (appendix). In addition, we added three more baseline LLMs, and the results show that UIS-Digger is a general framework that yields strong results and demonstrate a good regularity with different models. The improvement of UIS-Digger on the UIS-QA benchmark does not come from a specially chosen backbone model, but from the reasonable multi-agent system design.

5. Reviewer **xVJL** further criticized our novelty and asked about the differences between UIS-QA and existing benchmarks such as WebArena and Mind2Web. We regret that **xVJL** might not have fully captured some critical parts of our paper that define UIS problems. In our reply to ​**xVJL**​, we provided a detailed comparison table showing that UIS-QA is the first information-seeking benchmark focusing on unindexed information, which is fundamentally different from browser-operating benchmarks such as WebArena and Mind2Web. This point has been recognized by the other reviewers: **RP5P** comments on UIS-QA as "an interesting task for Unindexed Information Seeking (UIS)"; **9Jka** writes "This work identifies an interesting UIS problem, which is practical in reality and still under-explored"; and **oSEj** comments: "It identifies a novel and important gap that existing benchmarks overlook".

6. We also reported experiments showing that UIS-QA is not a trivial variant of existing benchmarks: even advanced DeepResearch systems, such as OpenAI-DR, achieve only about 20% accuracy. Beyond the benchmark itself, our UIS-Digger agent introduces a dual-mode browser with two complementary backbones (a strong task performing LLM and a VLM for visual perception) that share a unified chat history and webpage state, allowing the agent to actively choose which backbone to invoke at each step without losing context. To the best of our knowledge, existing text-only or multimodal browser frameworks do not support such a design. These points, detailed in our responses, further strengthen the contribution of our work, and we sincerely hope you could reconsider the low-score review from **xVJL** in light of the results and the other reviewers’ positive assessments.

To avoid being wordy, we do not repeat all details here. We kindly ask you to review our full responses to the reviewers. It is a pity that the rebuttal and discussion stage ended abruptly due to the unexpected system bug, but we are very much looking forward to your meta-review and final decision. Thanks so much for your time and consideration.

---

> ### Author Response · Authors · 2025-12-03
> **Briefly highlight the core contributions**
>
> Dear AC, as the discussion phase concludes, we would like to briefly highlight the core contributions of our work, including those that several reviewers have explicitly recognized.
>
> Our work is the first to identify and formulate **Unindexed Information Seeking (UIS)**, where crucial information is *not* captured by search engines (e.g., dynamic pages, embedded files). To provide a systematic and quantitative measurement in this new area, we introduce **UIS-QA**, the first information-seeking benchmark explicitly targeting unindexed information. Even strong agents that perform well on existing benchmarks suffer drastic drops on UIS-QA, for example, from 70.90 on GAIA and 46.70 on BrowseComp-zh to 24.55 on UIS-QA. Advanced proprietary systems also struggle: OpenAI Deep Research reaches only about 20%, and an advanced open-source system with a strong backbone model (o3 + Memento) achieves merely 25.5%, indicating that UIS-QA exposes a genuinely hard and previously neglected capability gap.
>
> To address this, we propose **UIS-Digger**, which (i) introduces a systematic UIS data synthesis framework, (ii) designs a **dual-mode browser workflow** for unified textual–visual interaction, and (iii) trains a ~30B-parameter model that already outperforms o3 + Memento on UIS-QA. Despite being designed for UIS, UIS-Digger also shows strong generalization to BrowseComp-zh, GAIA, and FinSearchComp, with consistent gains after SFT and RFT. We thus not only identify a previously neglected **new problem** but also provide a **benchmark dataset and a strong baseline system**, demonstrating that UIS problems are feasible and measurable, yet still challenging and calling for more carefully crafted technical designs. We believe this work could serve as a solid inspiration for future research on the UIS problem.

---

### Meta-Review · Area_Chair_4gAs · 2026-01-09

**Summary:**

Reviewers generally agree that the paper identifies and formulates the problem of Unindexed Information Seeking (UIS), introduces a new benchmark (UIS-QA), and presents a multi-agent baseline (UIS-Digger) with dual-mode browsing and file parsing **[RP5P, xVJL, 9Jka, oSEj]**.
Several reviewers view the identification of UIS and the effort to construct a carefully filtered benchmark as valuable and underexplored **[9Jka, oSEj, RP5P]**.
Empirically, reviewers acknowledge that agents experience large performance drops on UIS-QA and that UIS-Digger achieves the SOTA results on this benchmark **[RP5P, 9Jka, oSEj]**.

At the same time, reviewers raise concerns about the conceptual distinctness of UIS relative to existing web agent and information seeking benchmarks, the limited scale of UIS-QA, missing baselines, and the modest absolute performance of UIS-Digger.
While the rebuttal adds extensive clarification, new baselines, human evaluation, and additional statistics, reviewer opinions remain mixed **[xVJL]**

**Reviewer Concerns:**

**Addressed**
- Dataset scale and statistics: RP5P and 9Jka questioned size and coverage. The rebuttal adds human performance, domain/type statistics, and justification of scale.
- Missing baselines and generalization: RP5P and xVJL requested stronger baseline. The rebuttal adds OpenAI Deep Research and evaluations on GAIA, BrowseComp-zh, and FinSearchComp.
- Experimental detail and reproducibility: RP5P requested clearer SFT/RFT details. The rebuttal expands methodological descriptions and prompts

**Outstanding**
- Conceptual novelty of UIS: xVJL maintains that UIS largely overlaps with existing web agent benchmarks and reflects curation rather than a new problem
- Technical novelty of UIS-Digger: xVJL questions whether the architecture and training pipeline go beyond standard compnoents.
- Absolute performance and practical impact: 9Jka and oSEj note that accuracy remains low
- Breadth of UIS coverage: 9Jka questions whether UIS-QA fully captures the space of unindexed information scenarios.

**Reviewer Scores:**

Reviewer RP5P: Likely unchanged. The rebuttal addresses most requests, but they already indicate only marginal support and would not mind rejection.

Reviewer xVJL: Unchanged. They maintain a reject recommendation based on novelty and contribution.

Reviewer 9Jka: Likely unchanged. Clarifications help, but concerns about benchmark characterization and modest gains remain.

Reviewer oSEj: Likely unchanged. They are positive on the problem but continue to note low absolute performance.

---

### Decision · Program_Chairs · 2026-01-26

Accept (Poster)